# Impact of Atmospheric Rivers on Arctic Sea Ice Variations

Linghan Li[1], Forest Cannon[1], Matthew R. Mazloff[1], Aneesh C. Subramanian[2], Anna M. Wilson[1], F. Martin Ralph[1]

[1]Center for Western Weather and Water Extremes, Scripps Institution of Oceanography, University of California, San Diego, US
[2]Department of Atmospheric and Oceanic Sciences, University of Colorado Boulder, US

*Correspondence to*: Linghan Li (li.linghan.li@gmail.com)

**Abstract.** Arctic sea ice has been declining rapidly in recent decades. We investigate how poleward transport of moisture and heat from lower latitudes through atmospheric rivers (ARs) influences Arctic sea ice variations. We use ERA5 hourly reanalysis data for 1981-2020 at 0.25° x 0.25° resolution to examine meteorological conditions and sea ice changes associated with ARs in the Arctic. In years 2012 and 2020, which had extremely low summer Arctic sea ice extent, we show that individual AR events associated with large cyclones initiate rapid sea ice decrease through turbulent heat fluxes and winds. We further carry out statistical analysis of meteorological conditions and sea ice variations for 1981-2020 over the entire Arctic Ocean. We find that on weather timescales atmospheric moisture content anticorrelates significantly with sea ice concentration tendency almost everywhere in the Arctic Ocean, while dynamic sea ice motion driven by northward winds further reduces sea ice concentration.

## 1 Introduction

Atmospheric rivers (ARs) are long, narrow, and transient corridors of strong horizontal water vapor transport as defined in the Glossary of Meteorology (American Meteorological Society, 2017). ARs are an important form of extreme horizontal water vapor transport on weather and synoptic scales. ARs account for 90% poleward water vapor transport at midlatitudes (Zhu and Newell, 1998). Poleward water vapor transport by ARs plays a critical role in atmospheric moisture content and precipitation variations at high latitudes (e.g., Nash et al, 2018).

The Arctic has experienced rapid sea ice decrease especially in summer in recent decades. This decrease has been quantified using satellite observations since 1979, with the record low summer sea ice minimum occurring in 2012 (Parkinson and Comiso, 2013). Rapid Arctic sea ice loss leads to global climate change and weather extremes, such as increasing occurrence of ARs near the North American west coast (Ma et al., 2021). The imminence of the seasonal ice-free Arctic Ocean in the coming few decades (Peng et al., 2020; Notz et al., 2020; Guarino et al., 2020) has important implications for marine ecosystems, shipping, resources, and conservation.

Many processes and feedbacks contribute to sea ice variations in the Arctic across different temporal and spatial scales. Previous studies show that poleward atmospheric energy transport is a key factor driving Arctic sea ice variations (e.g. Olonscheck et al., 2019; Hofsteenge et al., 2022). In particular, moisture transport into the Arctic influence the surface energy budget by strengthening downward longwave radiation and sensible heat flux,

especially in winter (Doyle et al., 2011; Mortin et al., 2016; Woods and Caballero, 2016; Johansson et al, 2017; Woods et al., 2017; Hegyi et al., 2018; Olonscheck et al., 2019; Ali and Pithan, 2020; Wang et al., 2020; Fearon et al., 2020). Those studies are mostly on large spatial scales such as considering zonal mean moisture transport (e.g. Papritz et al., 2022; Hofsteenge et al., 2022) and total Arctic sea ice loss (e.g. Wernli and Papritz, 2018) and over long time scales such as interannual variability (e.g. Olonscheck et al., 2019) and trends (e.g. Woods and Caballero, 2016). ARs are one form of extreme moisture transport on weather and synoptic scales at low levels. However, the direct impact of ARs through both thermodynamic and dynamic processes on Arctic sea ice variations at high temporal and spatial resolutions still needs better understanding (Hegyi et al., 2018; Wang et al., 2020; Papritz et al., 2022), which is a primary goal of this paper.

ARs transport moisture and heat from lower latitudes into the Arctic with considerable potential to drive sea ice reduction (e.g., Baggett et al., 2016; Hegyi et al., 2018; Vázquez et al., 2018; Wang et al., 2020; Papritz et al., 2022). Though the upstream moisture source and transport pathways of ARs reaching the Arctic have been examined (Vázquez et al., 2019; Harrington et al., 2021; Papritz et al., 2022), the downstream interaction between ARs and sea ice deserves more studies. As low-level jets associated with ARs come in close contact with sea ice surface near ice margins, they can have direct impacts on sea ice through intense surface energy, momentum, and mass exchanges between the atmosphere and ice/ocean (Hegyi et al., 2018; Wang et al., 2020). Furthermore, the warm and moist air masses brought by ARs are forced upward by cold air to form precipitation and clouds with associated latent heat release into the upper air (e.g., Komatsu et al., 2018).

This study investigates the relative contribution of surface heat flux components and the relative importance of thermodynamic and dynamic processes in sea ice changes when ARs happen in the Arctic. Originality of this study includes revealing the important roles of turbulent heat fluxes under ARs and ARs' wind effect on sea ice change on weather timescales. Another novel aspect of this study is the method of extraction of high-frequency signals representing extreme weather events such as ARs, without mean seasonal cycle, interannual variability and trends. Furthermore, we find that the covariation between ARs and sea ice holds for the whole Arctic Ocean through the entire seasonal cycle. This comprehensive analysis provides better understanding of physical processes governing the interactions between ARs and sea ice with potential applications to short term sea ice prediction.

The goal of this study is to explore how ARs influence Arctic sea ice variations. We examine sea ice changes in relation to AR forcing by thermodynamic surface heat fluxes and dynamic winds. First, we show two case studies to examine physical processes through spatial pattern and time series analysis. Second, we carry out statistical analysis for 1981-2020 over the entire Arctic Ocean.

**2 Data and Methods**

ERA5, the fifth-generation European Re-Analysis, is the most recent atmospheric reanalysis from ECMWF, with data assimilation system IFS Cycle Cy41r2 with 4D-Var (Hersbach et al., 2020). ERA5 uses boundary conditions of sea ice concentration and sea surface temperature based on satellite observations (HADISST2 before 1979, OSI SAF (409a) for 1979-Aug 2007, OSI SAF operational from Sep 2007). ERA5 has high temporal (hourly) and spatial (~31km horizontally, 137 model levels vertically) resolutions suitable for studying extreme weather events such as ARs. Its long and consistent records are also appropriate for studying climate change and variability.

ERA5 atmospheric reanalysis over Arctic sea ice has been evaluated with in-situ observations (Graham et al., 2019; Batrak et al., 2019; Renfrew et al., 2020). ERA5 accurately represents integrated water vapor (IWV) and downward longwave radiation, and correlations between ERA5 and in-situ observations are high for meteorological variables such as IWV, 2-m temperature, 10-m wind speed, and sea level pressure (Graham et al., 2019). Noticeably, ERA5 improves estimate of the Arctic energy budget in terms of closure (Mayer et al., 2019). However, ERA5 has a warm bias near the Arctic sea ice surface, which is large in winter and small in summer (Graham et al., 2019; Batrak et al., 2019).

We use ERA5 hourly reanalysis data for 1981-2020 at 0.25º x 0.25º resolution in the Arctic. First of all, we study two AR events associated with cyclones during summertime of 2012 and 2020, which are periods of record low Arctic sea ice extent. We examine spatial patterns and time series of meteorological conditions, surface energy budget, and sea ice variations related to the AR events. Furthermore, for each grid box of the entire Arctic Ocean, we investigate the statistical relationship between anomalous meteorological conditions and sea ice variations on weather timescales during 1981-2020. We extract high frequency variations on weather timescales and identify extreme moisture anomalies as approximate ARs which are validated by Guan and Waliser's AR catalog version 3 from 6-hourly 0.625° x 0.5° Modern-Era Retrospective Analysis for Research and Applications, Version 2 (MERRA-2) reanalysis data for 1980-2020 (Guan and Waliser, 2019).

**3 Results**

**3.1 Two Case Studies in 2012 and 2020**

Atmospheric rivers transport moisture and heat from lower latitudes into the Arctic at low levels on weather timescales. Those episodic extreme events of water vapor transport can have a large impact on Arctic sea ice variations with important implications for short-term sea ice prediction. Here we show two examples of AR events occurring in the summertime of 2012 and 2020 in the western Arctic Ocean associated with rapid sea ice changes through turbulent heat fluxes, longwave radiation, and winds. These case studies of AR events reveal important physical processes and give context to the statistical analysis shown later.

**3.1.1 The Atmospheric River Event of August 2012**

In September 2012 the Arctic sea ice extent reached the lowest ever observed since satellite observations started in 1979 (Parkinson and Comiso, 2013). Before that, an Arctic cyclone with the lowest central pressure (966 hPa) recorded in August since 1979 occurred on August 5-12, 2012 (Simmonds and Rudeva, 2012). On August 5, 2012, when the cyclone first reached the Arctic Ocean, an associated AR entered the western Arctic Ocean.

Sea ice concentration decreases substantially in the Chukchi Sea and the East Siberian Sea between Aug 4-6, 2012 (Figure 1a). The AR enters the western Arctic from Siberia on Aug 4, 2012, reaches its highest intensity on Aug 5, 2012, with integrated water vapor transport (IVT) values exceeding 250 kg m$^{-1}$ s$^{-1}$ (maximum in the core around 500 kg m$^{-1}$ s$^{-1}$), and is still present in the Arctic Ocean on Aug 6, 2012 (exceeding 250 kg m$^{-1}$ s$^{-1}$) (Figure 1b). Concurrently, strong northeastward surface winds (inferred from sea level pressure) beneath the AR core push sea ice away from the ice edge towards the pole. Coincident with the AR in space and time, strong downward sensible and latent heat flux happens around the ice edge, due to warm and moist air and high wind speed at low levels within the AR (Figure 1cd). Net longwave radiation (incoming longwave radiation minus outgoing longwave radiation) near the ice edge is also downward, but the magnitude is much weaker than turbulent heat fluxes (Figure 1f). In contrast, net shortwave radiation (incoming shortwave radiation minus outgoing shortwave radiation) is reduced over sea ice cover (Figure 1e). The radiation patterns, reduced net shortwave radiation and enhanced net longwave radiation, are consistent with clouds and precipitation that formed when the AR arrived (not shown). This major event gives evidence that turbulent heat fluxes can be the dominant terms in surface energy budget during ARs. In contrast, most former studies emphasize dominant downwelling longwave radiation related to water vapor and clouds (Doyle et al., 2011; Mortin et al., 2016; Woods and Caballero, 2016; Johansson et al, 2017; Woods et al., 2017; Hegyi et al., 2018; Wang et al., 2020), though some studies reveal an important role of surface sensible heat flux (e.g., Stern et al., 2020).

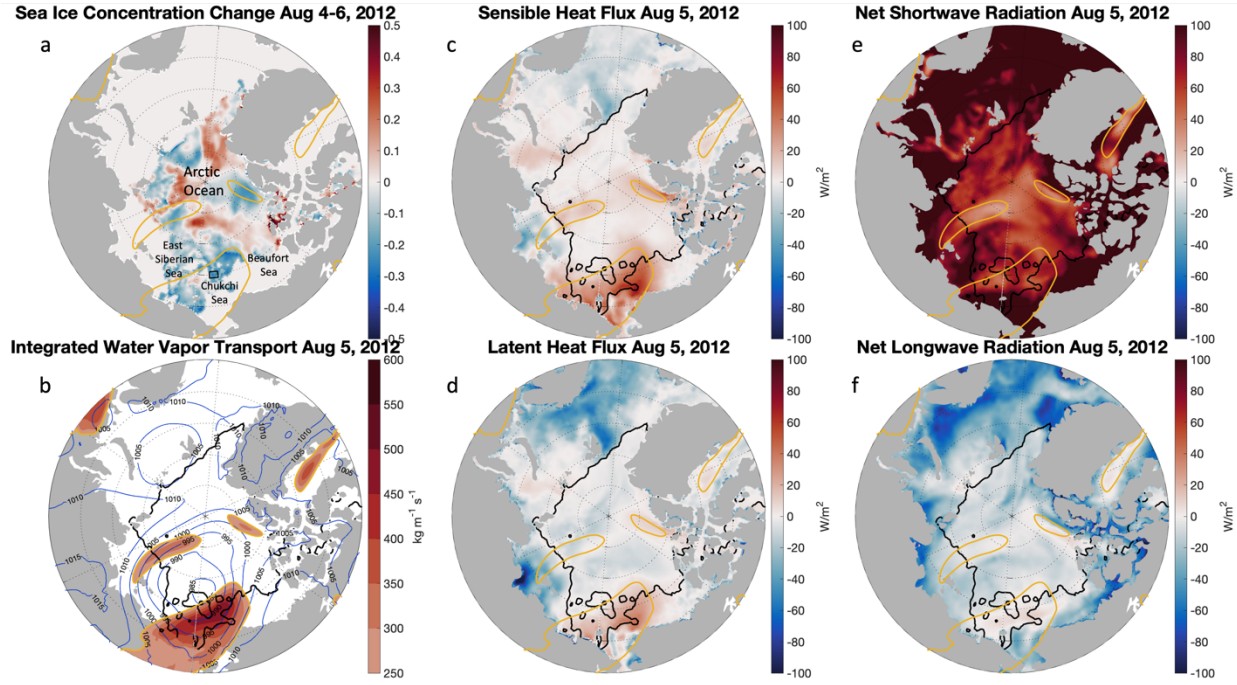

**Figure 1: a** Sea ice concentration change during Aug 4-6, 2012 (sea ice concentration on Aug 6, 2012 minus sea ice concentration on Aug 4, 2012). Extent of atmospheric rivers (250 kg m$^{-1}$ s$^{-1}$ contour of integrated water vapor transport) on Aug 5, 2012 is represented as yellow lines (same in b c d e f). **b** Magnitude of integrated water vapor transport on Aug 5, 2012. Sea ice edge (15% contour of sea ice concentration) on Aug 5, 2012 is represented as black lines (same in c d e f). Sea level pressure on Aug 5, 2012 is represented as blue contours. **c** Sensible heat flux (positive downward) on Aug 5, 2012. **d** Latent heat flux (positive downward) on Aug 5, 2012. **e** Net shortwave radiation (positive downward) on Aug 5, 2012. **f** Net longwave radiation (positive downward) on Aug 5, 2012. Time is in UTC.

Figure 1 shows the original fields to demonstrate the magnitude of the event. However, the seasonal cycle can impact the interpretation of the event, as radiative and turbulent fluxes change their relative importance throughout the year. We therefore further investigate this event using anomalies (Figure 2). Anomaly is defined as difference between original field and the daily 1981-2020 climatology for the case study here. Indeed, Figure 2 removes the mean radiative component features, and doesn't reveal significantly large radiative anomalies collocated with anomalous IVT. As expected, shortwave is reduced and longwave is enhanced in regions of anomalous IVT, but the magnitudes are similar to anomalies found elsewhere in the region. The most significant anomalous flux component collocated with IVT is sensible heat flux. This positive flux anomaly is augmented by additional anomalous heating from latent and longwave components. However, like shortwave anomalies, longwave anomalies associated with increased IVT are not as notable in the domain. Figure 2 makes clear that turbulent flux anomalies in the Chukchi Sea are notable during this event and collocate with anomalous IVT, whereas significant radiative anomalies can be found throughout the domain especially with regards to the shortwave component.

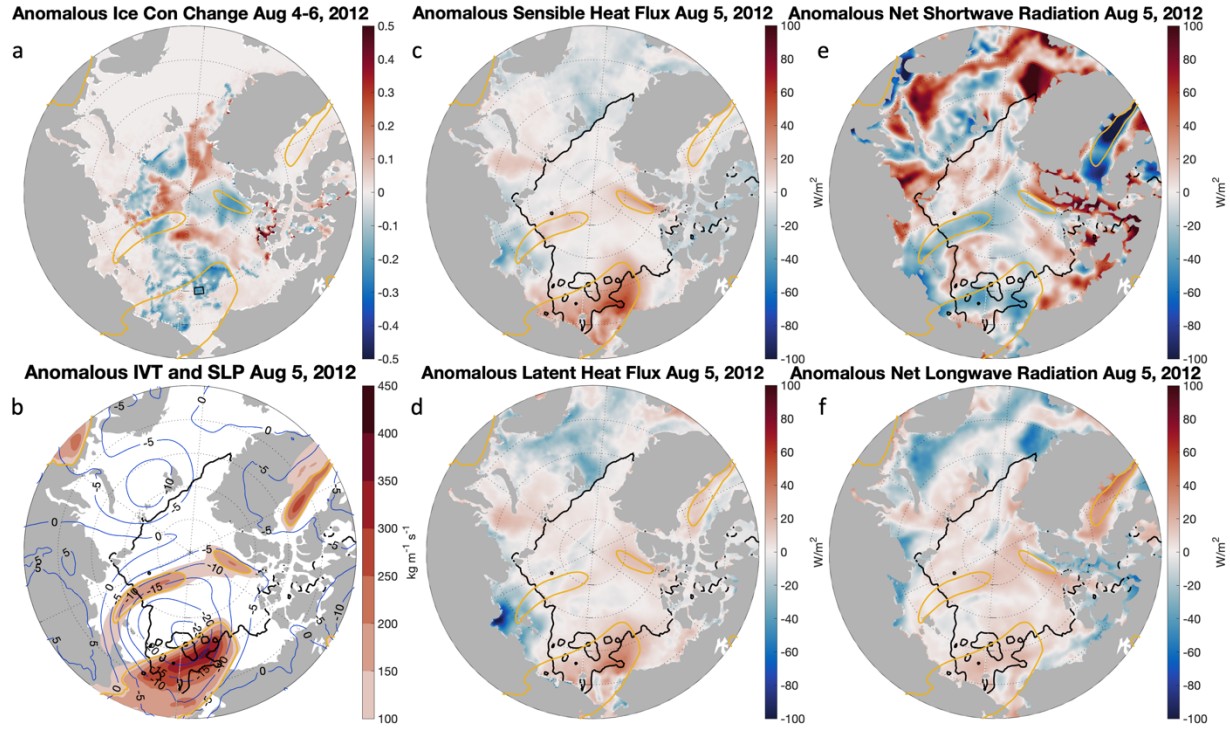

**Figure 2: Same as Figure 1 but using anomalous fields (original fields minus climatologies) on Aug 5 (Aug 4-6 for ice concentration change) during 1981-2020. Note that sea ice edge in black lines and extent of atmospheric rivers in yellow lines are based on original fields.**

In order to study the total effect of the cyclone on sea ice, we also examine time integrated surface heat fluxes, time integrated IVT, and sea ice concentration change through the life cycle of the cyclone (Supplementary Figure S1). During Aug 5-12, 2012, sea ice cover is reduced substantially over broad areas of the Chukchi Sea and the East Siberian Sea, corresponding with strong time integrated IVT there (maximum around 1.5 kg m$^{-1}$ over 7 days). Strong sensible heating, weak latent heating, weak net longwave radiation cooling, and weak net shortwave radiation heating are seen there. The effect of this cyclone on sea ice during its life cycle is extensive sea ice decrease in the western Arctic Ocean, corresponding to strong IVT, atmospheric warming (enhanced sensible/latent heat fluxes and longwave radiation) and sea ice motion driven by winds. Here we only consider local atmospheric forcing of sea ice changes during the cyclone, though other processes such as ocean-ice heat flux are also important in sea ice melt during the cyclone (Zhang et al., 2013; Stern et al., 2020; Finocchio et al., 2020; Lukovich et al, 2021). Though we focus on ARs instead of cyclones in this study, the roles of cyclones/anticyclones in sea ice melt are very complicated. For example, enhanced Arctic summer sea ice melt is related to polar anticyclones and extratropical cyclones (Wernli and Papritz, 2018). Cyclones seem to have little effect on sea ice (Hofsteenge et al., 2022).

We examine hourly time series of surface heat fluxes and meteorological conditions and daily time series of sea ice concentration and sea surface temperature for Aug 1-15, 2012 averaged over the black box as shown in Figure 1a

(Figure 3). The location of the black box is chosen due to its proximity to the summer ice edge in the western Arctic Ocean. This study area was partially covered by sea ice before the cyclone and became ice free after the cyclone. Sea ice concentration and sea surface temperature from ERA5 are daily based on satellite observations, while other variables from ERA5 are hourly in Figure 3. The time series shows that sea ice concentration drops abruptly around Aug 5, 2012 when the AR arrives (Figure 3a). Correspondingly, large downward sensible and latent heat fluxes and

moderate downward net longwave radiation into ice/ocean occurs within one day, peaking at midnight with nearly zero net shortwave radiation on Aug 5, 2012 (Figure 3b). The dominant latent and sensible heat fluxes are related to the high moisture and heat content associated with the AR, while high wind speed further enhances turbulent heat fluxes. This AR event on Aug 5, 2012 over the black box is categorized as an AR Cat 2 (Moderate) with maximum IVT magnitude 867 kg m$^{-1}$ s$^{-1}$ and duration of 18 hours (IVT>250 kg m$^{-1}$ s$^{-1}$) (AR scale is defined by Ralph et al.,

2019). We further partition IVT into moisture and wind components to separate thermodynamics and dynamics of ARs. The specific humidity at 850 hPa shows a prominent peak on Aug 5, 2012 (Figure 3c). The 850 hPa wind speed is the strongest at the same time as the largest specific humidity on Aug 5 and continues to be strong for the next 3 days (Figure 3c). Wind direction is northward on Aug 5, 2012, and then turns eastward on Aug 6-10, 2012 (not shown). Near surface conditions in humidity and wind are also examined and show similar results (not shown).

In summary, simultaneous peaks in moisture and wind speed cause intense downward turbulent heat fluxes and subsequent rapid sea ice decrease when the AR arrives on Aug 5, 2012.

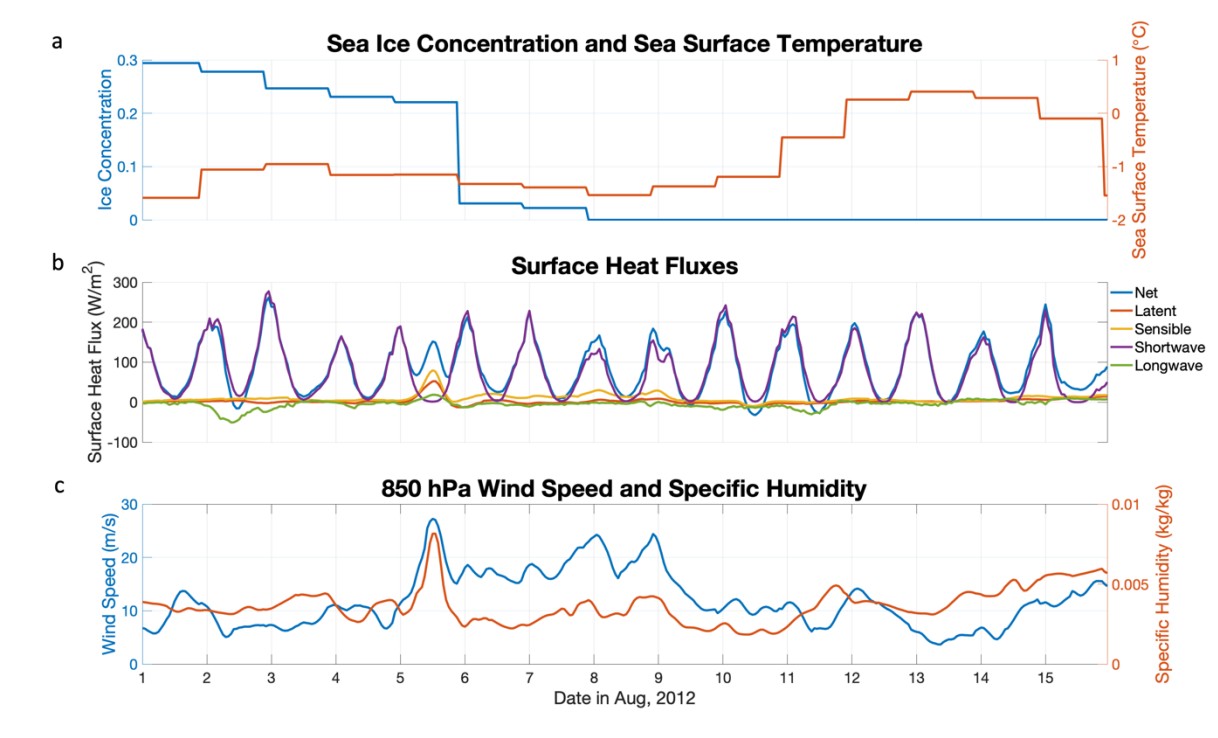

**Figure 3: a Sea ice concentration (blue) and sea surface temperature (red) daily during Aug 1-15, 2012 averaged over the**

**black box in Figure 1 a. b Latent heat flux (red), sensible heat flux (yellow), net shortwave radiation (purple), net longwave radiation (green), and net surface heat flux (blue) hourly during Aug 1-15, 2012 averaged over the black box in**

**Figure 1 a. c Wind speed (blue) and specific humidity (red) at 850 hPa hourly during Aug 1-15, 2012 averaged over the black box in Figure 1 a. Time is in UTC, and dates marked on time axis represent 00:00 UTC.**

Sea surface temperature at this location decreases slightly after the AR during the first half of the cyclone and increases substantially near the end of the cyclone (Figure 3a). The spatial pattern of sea surface temperature change during the life cycle of the cyclone reveals significant ocean warming in newly formed open ocean which was covered by sea ice before the cyclone (Supplementary Figure S2). This ocean surface warming is related to strong air-sea interaction during the cyclone with shortwave radiation as the dominant term in surface heat fluxes. Wind

induced ocean mixing could also bring subsurface warm water upward during the cyclone (Zhang et al., 2013; Stern et al., 2020).

**3.1.2 The Atmospheric River Event of July 2020**

We study another extreme event in summer of 2020. 2020 experienced the second lowest summer sea ice extent and the lowest sea ice extent during spring, early summer and fall in the Arctic, based on the NSIDC sea ice index (Fetterer et al., 2017). The summer 2020 sea ice deficit occurred largely in the western Arctic Ocean (Liang et al., 2022). The reasons causing the record low Arctic sea ice in 2020 involve several processes across different temporal and spatial scales. The Siberian heat wave associated with local atmospheric warming occurred during Jan-June

2020 (Overland and Wang, 2020) before the extremely low summer sea ice extent in the East Siberian Sea. The record low Arctic sea ice extent in July, 2020 has also been attributed to horizontal transport of heat and moisture during April-June, 2020 (Liang et al., 2022). Here we show an example of the contribution of strong water vapor transport by ARs on weather timescales to rapid sea ice changes during summer of 2020.

In the western Arctic Ocean, a large cyclone occurred during July 25 - Aug 2, 2020, reaching lowest central pressure (968 hPa) on July 28, 2020. Strong advection of heat and moisture (with large gradient of temperature and moisture) from Siberia to the Arctic Ocean occurred on July 26, 2020 (not shown). An AR event with maximum IVT around 600 kg m$^{-1}$ s$^{-1}$ in the core and strong northward winds happened near the ice edge on July 27, 2020 (Figure 4b). Corresponding to the AR timing and location over sea ice, strong sensible/latent heat fluxes and net longwave

radiation are directed from the atmosphere into the ice/ocean, and net shortwave radiation is reduced by clouds (Figure 4cdef). Meanwhile, strong northeastward winds reduce sea ice concentration near the ice edge (Figure 4ab). The wind direction is northward before the AR and becomes eastward after the AR (not shown). This suggests wind-driven meridional water vapor transport at the early stage of the cyclone life cycle and intensified IVT as the AR propagates eastward along the coast later.


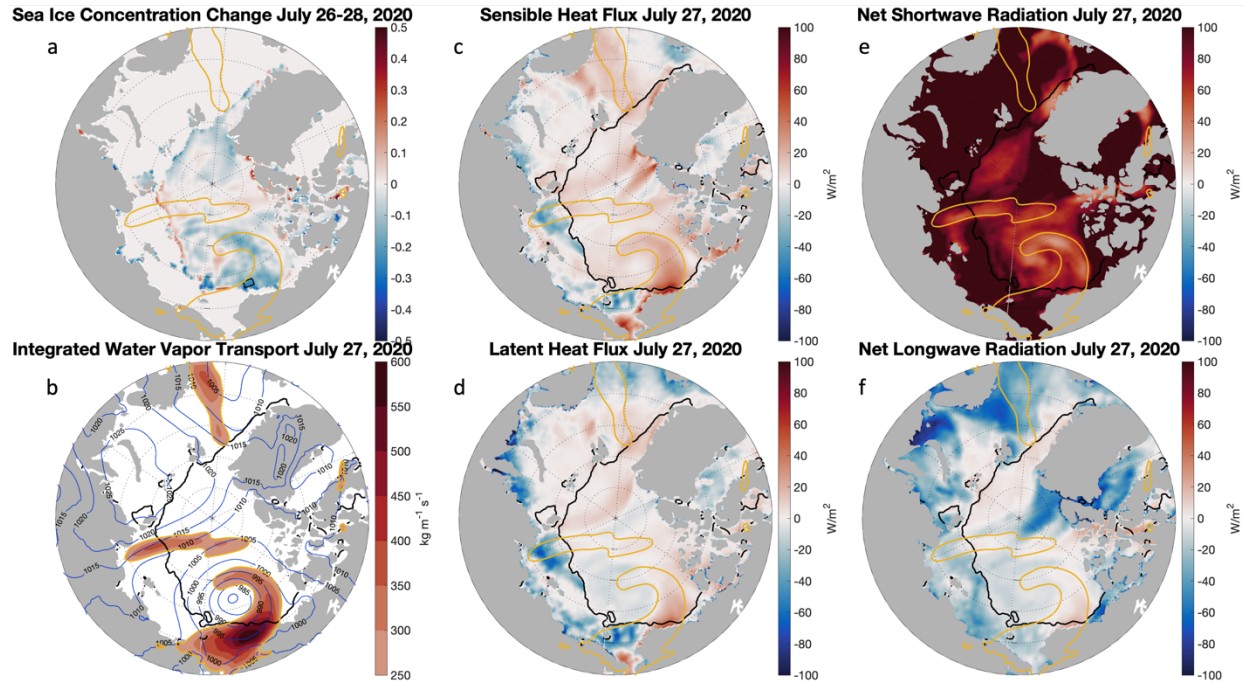

**Figure 4: a** Sea ice concentration change during July 26-28, 2020 (sea ice concentration on July 28, 2020 minus sea ice concentration on July 26, 2020). Extent of atmospheric rivers (250 kg m$^{-1}$ s$^{-1}$ contour of integrated water vapor transport) on July 27, 2020 is represented as yellow lines (same in b c d e f). **b** Magnitude of integrated water vapor transport on July 27, 2020. Sea ice edge (15% contour of sea ice concentration) on July 27, 2020 is represented as black lines (same in c d e f). Sea level pressure on July 27, 2020 is represented as blue contours. **c** Sensible heat flux (positive downward) on July 27, 2020. **d** Latent heat flux (positive downward) on July 27, 2020. **e** Net shortwave radiation (positive downward) on July 27, 2020. **f** Net longwave radiation (positive downward) on July 27, 2020. Time is in UTC.

As in Figure 2, we also investigate this event with respect to difference from the daily 1981-2020 climatology. This highlights the spatial extent of anomalies of surface heat fluxes are collocated well with anomalous IVT (Figure 5). Like the event discussed earlier, the most striking anomalous fluxes are the turbulent fluxes: the sensible heat flux is greatest where the IVT is greatest, and the latent heat flux is negative over much of the domain, but strongly positive over the locations of high anomalous IVT. The radiative fluxes show the influence of the IVT anomaly, but these flux anomalies are large in many parts of the domain.

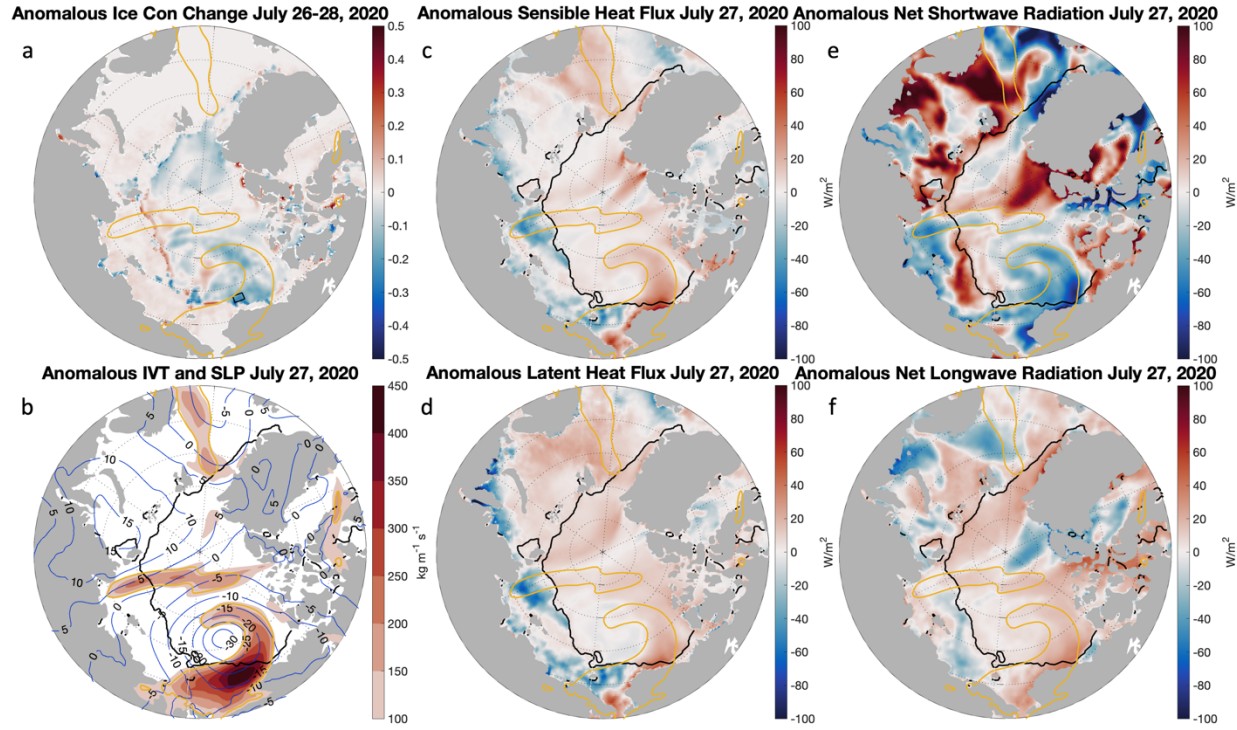

**Figure 5: Same as Figure 4 but using anomalous fields as departures from climatology on July 27 (July 26-28 for ice concentration change) during 1981-2020. Note that sea ice edge in black lines and extent of atmospheric rivers in yellow lines are based on original fields.**

Next, we examine time series of sea ice concentration, sea surface temperature, surface heat fluxes and AR conditions during the life cycle of the cyclone averaged over the black box as shown in Figure 4a (Figure 6). We choose this study area in the Beaufort Sea because it experiences the most significant sea ice reduction during the life cycle of the cyclone (July 25-Aug 2, 2020). At this location, sea ice concentration decreases gradually from 72% to 19% in one week throughout the cyclone, while the most rapid sea ice decrease happens immediately after the AR. Sea surface temperature increases immediately after the AR and persistently during the cyclone, indicating increasing ocean-ice heat flux for bottom and lateral sea ice melt. Culminating downward turbulent heat fluxes and weak downward net longwave radiation occur on early July 27, 2020 when the AR arrives. For this AR event on July 27, 2020, peaking moisture content, along with high wind speed, generates peaking downward turbulent heat fluxes. This AR event is categorized as AR 1 Scale (Ralph et al., 2019) with maximum IVT magnitude 502 kg m$^{-1}$ s$^{-1}$ and duration of 16 hours (IVT>250 kg m$^{-1}$ s$^{-1}$) (not shown). The major difference between this case and the first case in Aug 2012 is that sea ice concentration decreases gradually, lasting for a few days during the cyclone, which might be related to ocean's role due to warming of sea surface temperature and constant high wind speed during the cyclone.

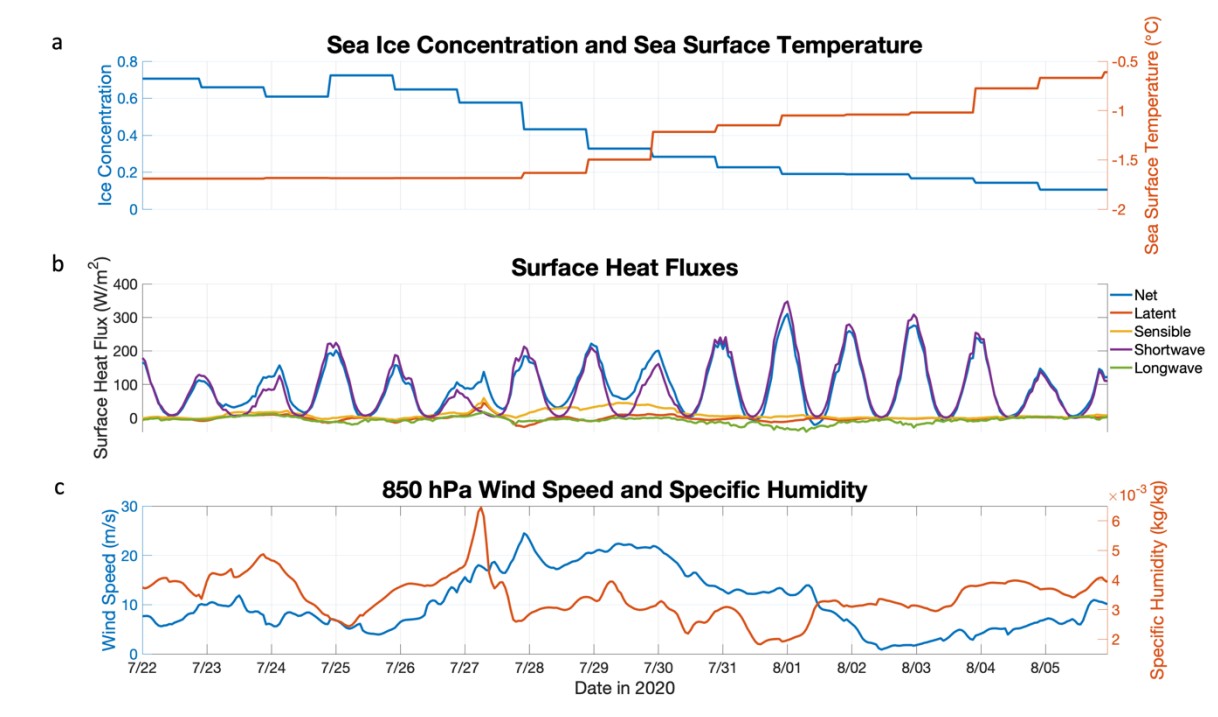

**Figure 6: a** Sea ice concentration (blue) and sea surface temperature (red) hourly during July 22-Aug 5, 2020 averaged over the black box in Figure 4 a. **b** Latent heat flux (red), sensible heat flux (yellow), net shortwave radiation (purple), net longwave radiation (green), and net surface heat flux (blue) hourly during July 22-Aug 5, 2020 averaged over the black box in Figure 4 a. **c** Wind speed (blue) and specific humidity (red) at 850 hPa hourly during July 22-Aug 5, 2020 averaged over the black box in Figure 4 a. Time is in UTC, and dates marked on time axis represent 00:00 UTC.

Another extreme event with the 3[rd] lowest summer Arctic sea ice minimum in 2007 has also been attributed to atmospheric heat and moisture transport from the Pacific with enhanced downwelling longwave radiation and turbulent fluxes (Graversen et al., 2011).

In summary, for the years 2012 and 2020, which experienced record low summer sea ice extent in the Arctic Ocean since 1979, AR related atmospheric water vapor transport from lower latitudes triggered rapid sea ice melt in the western Arctic Ocean through turbulent heat fluxes. Our results of surface energy budget of sea ice are generally consistent with in situ observations (Tjernström et al., 2015; Tjernström et al., 2019) and coupled atmosphere/ocean/ice models (Stern et al., 2020) showing the dominant role of turbulent heat fluxes under ARs. Our future work will examine the relative contribution of water vapor/heat transport and local warming/moistening in sea ice decline in the Arctic.

**3.2 Statistical Analysis of Meteorological and Sea Ice Variations for 1981-2020**

While the case studies in the previous section reveal linkages between AR-related atmospheric forcing and rapid sea ice changes, here we extend to a more general study for 40 years over the entire Arctic Ocean. We carry out

statistical analyses to examine the relationships between meteorological conditions and sea ice changes in the Arctic

Ocean using ERA5 reanalysis for 1981-2020. We explicitly separate thermodynamic and dynamic effects of ARs on

sea ice changes. We consider IWV, surface latent and sensible heat fluxes as thermodynamic variables, near-surface

northward wind as a dynamic variable, and ice concentration tendency on a daily basis. Because ERA5 has good

performance of IWV (Graham et al., 2019) and ice concentration is daily based on satellite observations, it is

reasonable to examine the relationship between IWV and ice concentration tendency on the daily basis to study how

ARs influence sea ice changes thermodynamically. In addition, we use northward wind as the driving force to

approximately study ARs' dynamic effect on sea ice. The partial sea ice cover with ice concentration < 85% we

consider is in free drift with small internal ice stress (Heorton et al., 2019). Sea ice velocity in free drift is strongly

correlated with wind forcing especially on weather timescales. We find that ice concentration tendency has

significant anticorrelation with IWV, northward wind, and turbulent heat flux on weather timescales almost

everywhere in the Arctic Ocean.

### 3.2.1 Seasonal Variations in 2012 at One Location in the Chukchi Sea

To put the AR event on Aug 5, 2012 in the context of longer time scales, we first extend the time series to the full

calendar year 2012 for the same study area shown as the black box in Figure 1a. We examine 2012 daily IWV, ice

concentration tendency, latent heat flux, sensible heat flux, and northward wind averaged over that study area

(Figure 7). Since satellite observed ice concentration is available at daily resolution, we consider daily mean IWV,

latent/sensible heat fluxes, and northward wind. The conditions of the extreme event on Aug 5, 2012 are marked

with black circles (Figure 7). For this extreme event, substantial sea ice concentration decrease corresponds to large

atmospheric moisture content, strong latent and sensible heating from the atmosphere, and high northward wind.

Note that those variables show dominant seasonal cycles and high frequency fluctuations imposing on seasonal

cycles. Those high frequency variations on weather timescales are the focus of this study.

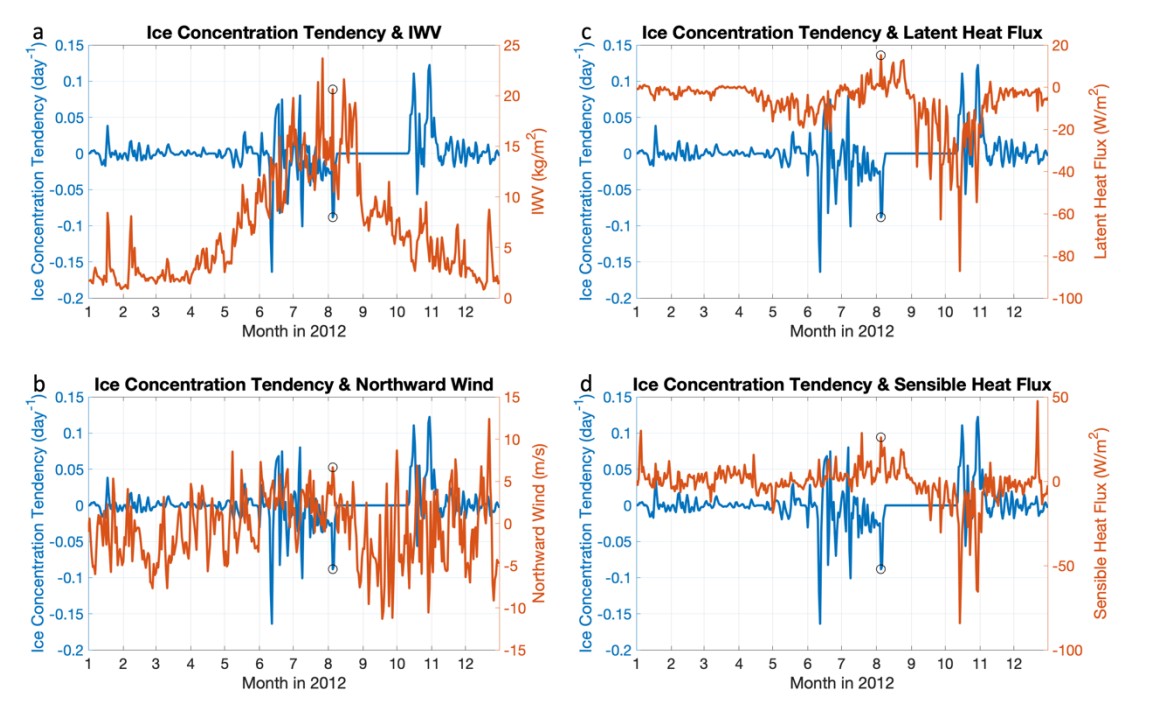

**Figure 7: a Variations in ice concentration tendency and IWV in 2012. b Variations in ice concentration tendency and northward wind in 2012. c Variations in ice concentration tendency and latent heat flux in 2012. d Variations in ice concentration tendency and sensible heat flux in 2012. All variables are averaged over the black box in Figure 1 a. The AR event on Aug 5, 2012 is marked with black circles.**

### 3.2.2 Statistical Analysis for 1981-2020 at One Location in the Chukchi Sea

We expand our time series analysis to 40 years for 1981-2020. We focus on variations on weather timescales in this study. We use a high pass filter with period <30 days to remove time-varying seasonal cycles from daily time series for 1981-2020. We define those high frequency variations (period <30 days) as anomalies in the statistical analysis here. We only consider time periods when ice cover is partial (ice concentration between 15% and 85%). This is in general consistent with the definition of marginal ice zone with ice concentration between 15% and 80%. Also, sea ice is in free drift for ice concentration less than 85%. The reason for this choice is that ice concentration can only change substantially in response to atmospheric forcing when ice concentration is partial.

We find significant rank correlation between anomalies of ice concentration tendency and IWV, northward wind, latent heat flux, and sensible heat flux for 1981-2020 (Figure 8). Rank correlation between ice concentration tendency and IWV anomalies is -0.30, and it is -0.28 for latent heat flux, -0.30 for sensible heat flux, and -0.34 for northward wind), all with p-values <0.01. The negative correlations are moderate, but significant due to the large sample size (14610). The big red dot in Figure 8a represents the AR event on Aug 5, 2012, which stands out as an extreme event during 1981-2020.

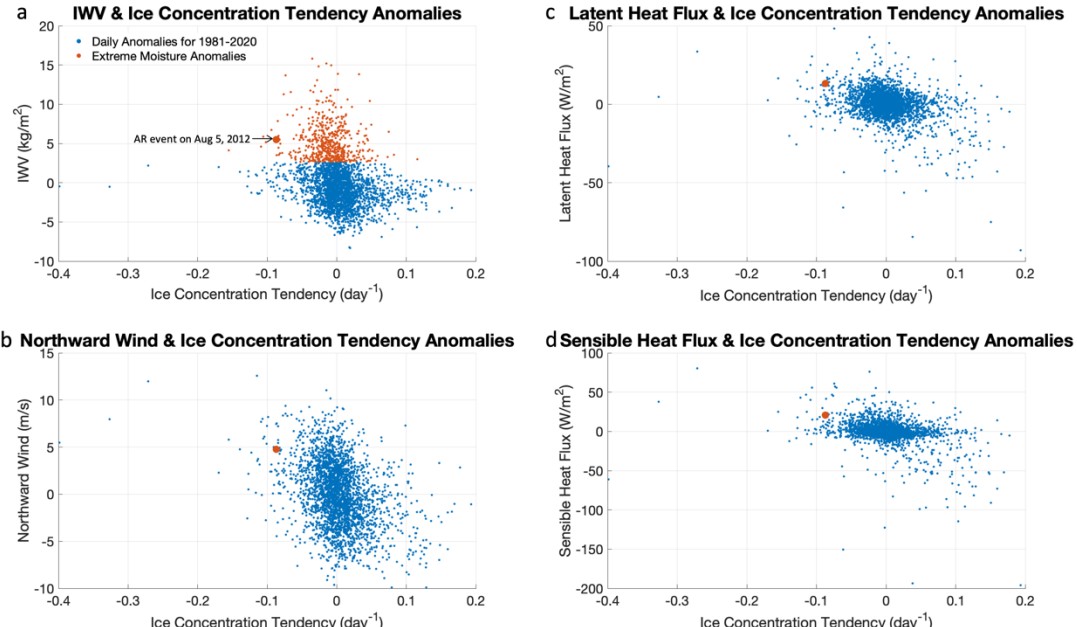

Figure 8: Scatter plots at one location in the Chukchi Sea (black box in Figure 1 a) from daily time series for 1981-2020 when ice cover is partial. a Scatter plot of anomalies of IWV and ice concentration tendency (blue dots), with small red dots representing extreme IWV anomalies exceeding 90% percentile. b Scatter plot of anomalies of northward wind and ice concentration tendency c Scatter plot of anomalies of latent heat flux and ice concentration tendency d Scatter plot of anomalies of sensible heat flux and ice concentration tendency. Big red dots in abcd represent the AR event on Aug 5, 2012.

Next, we identify all extreme moisture anomalies from ERA5 during 1981-2020 in this study area as proxies for AR events and validate them with a global AR catalog. We define extreme moisture anomalies as IWV anomalies exceeding the 90% percentile based on the daily time series during 1981-2020. We only consider extreme events over partially covered sea ice in this study. For this study area, we identify 553 extreme moisture anomalies with 20% frequency when sea ice cover is partial during 1981-2020. The mean IWV anomalies of those extreme events is 5.21 kg/m$^2$, and the mean ice concentration tendency anomalies of those extreme events is -1% day$^{-1}$. In Figure 8a, small red dots represent extreme moisture anomalies using ERA5 data, while the big red dot represents the AR event on Aug 5, 2012 (Figure 1 and Figure 3). We validate extreme moisture anomalies identified by our method with ARs identified by Guan and Waliser's AR catalog version 3 (Guan and Waliser, 2019). Among extreme moisture anomalies exceeding 90% percentile over partial sea ice cover, 72% have a corresponding AR identified from the catalog around the same date (+/-12h) at the same location. For 2012, this ratio is 91%. For large extreme events, these two methods agree very well. The consistency of results using methods based on IWV and IVT respectively implies that extreme IWV anomalies in the Arctic are due to water vapor transport by ARs from lower latitudes. In summary, good agreement of ARs identified by 2 methods indicates that it is reasonable to use extreme moisture

anomalies to approximate ARs in our study, but our method of detecting daily extreme moisture anomalies is simple and efficient for large datasets from ERA5. Additionally, our method of identifying extreme moisture anomalies is similar to moist-air intrusions defined in Papritz et al., 2022, except that we examine moisture evolution for each grid box accounting spatial variations while they consider zonal mean moisture transport (Papritz et al., 2022).


We further examine composite meteorological and ice conditions on dates with extreme IWV anomalies identified from 1981-2020 ERA5 data in the same study area. As shown in Figure 9a, a prominent feature of IWV is moisture intrusion onto sea ice cover over the Chukchi Sea. Simultaneously, a very strong northward wind pattern around the same region can transport water vapor from the Pacific to the Arctic (consistent with Baggett et al., 2016; Horvath et

al., 2021). Correspondingly, substantial sea ice loss occurs over a broad area where moisture reaches sea ice (Figure 9b). Those extreme events mainly happen during the summer season when sea ice cover is partial. We also examine composites on dates with extreme IWV anomalies and ARs from AR catalog. The spatial patterns of IWV, sea level pressure and ice concentration tendency composites with ARs are consistent with composites with extreme moisture anomalies, but the magnitudes are stronger (Supplementary Figure S3).


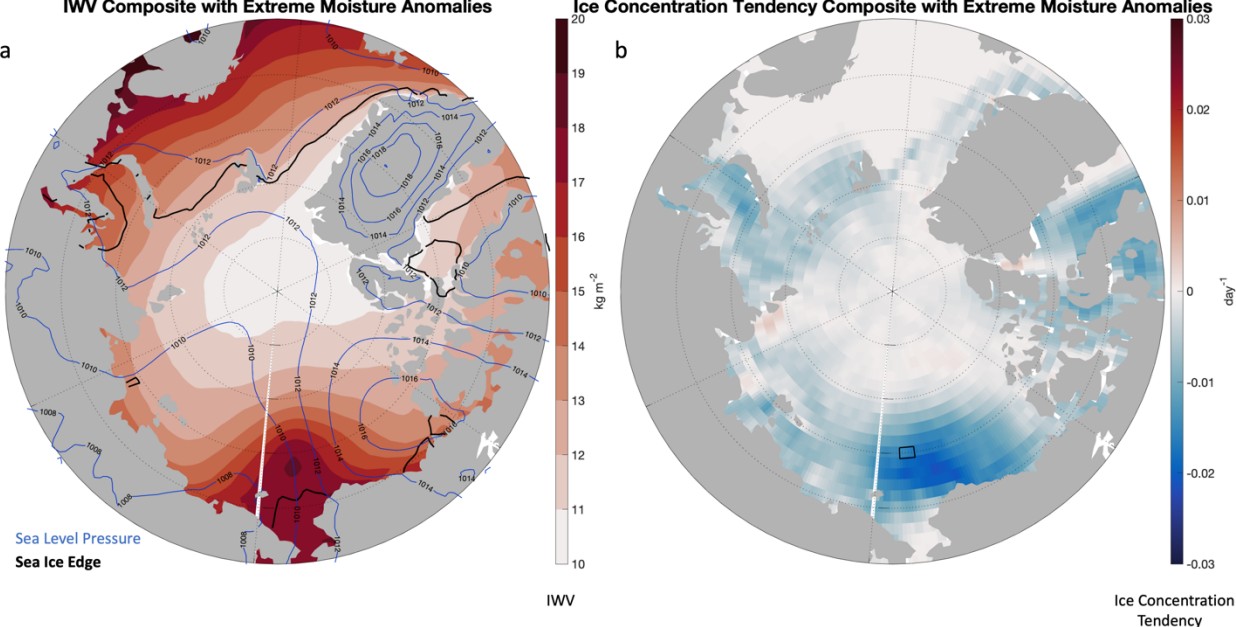

**Figure 9: a Composites of IWV, sea ice edge (black lines), and sea level pressure (blue contours) b Composite of ice concentration tendency when extreme IWV anomalies (>90% percentile) during 1981-2020 occur over partial ice cover in the black box in the Chukchi Sea.**


### 3.2.3 Statistical Analysis for 1981-2020 Everywhere in the Arctic Ocean

We further expand our analysis from one location in the Chukchi Sea to each grid box in the Arctic Ocean on the daily basis during 1981-2020. We find that anomalies of IWV and ice concentration tendency have significant

anticorrelation almost everywhere when sea ice cover is partial (Figure 10). Correspondingly, latent and sensible heat flux anomalies anticorrelate with ice concentration tendency anomalies. Dynamically, northward wind anomalies have negative correlation with ice tendency anomalies for the majority of the Arctic Ocean and have positive correlation with ice tendency anomalies near land margins, e.g., around Greenland. The synergy of thermodynamical and dynamical components of atmospheric forcing indicates the importance of horizontal

advection of heat and water vapor from lower latitudes through ARs in causing rapid sea ice changes. Note that only time periods with partial sea ice cover with ice concentration between 15% and 85% are included in the analysis, which are different at different locations. The marginal ice zone advances and retreats across different seasons. Thus, we consider different seasons for different regions. For instance, correlations are during the summer season in the central Arctic, while correlations are during the winter season in the southeast Bering Sea.


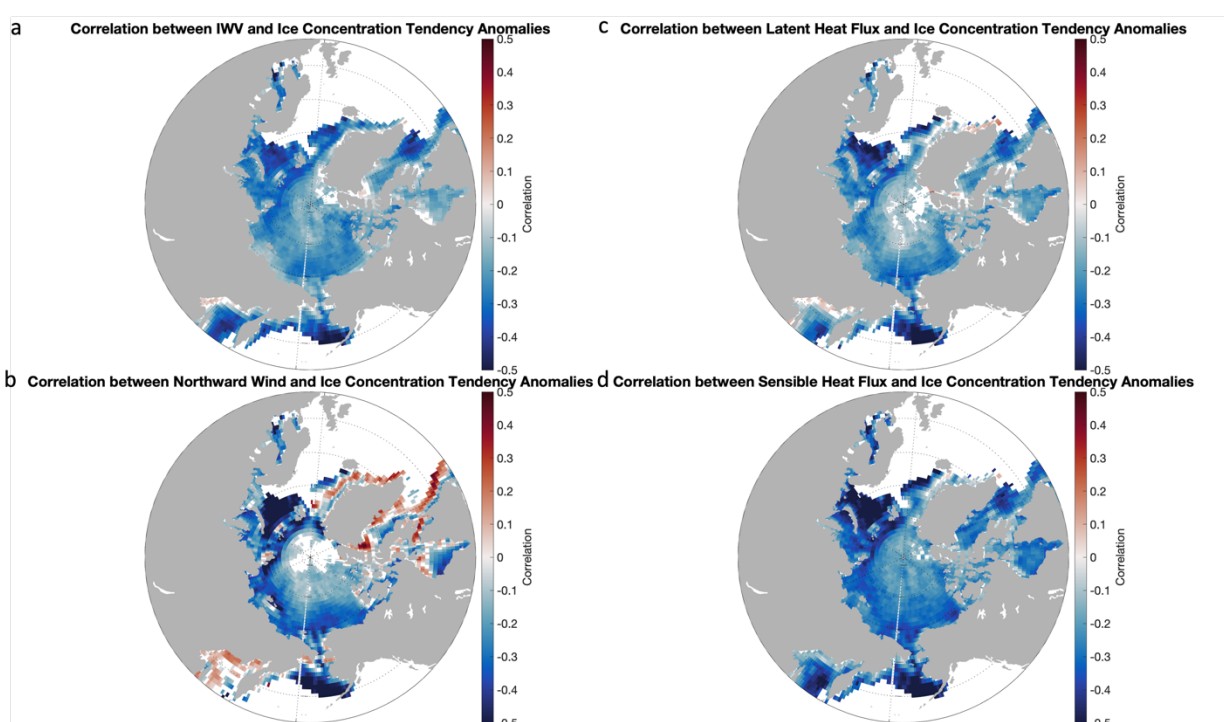

**Figure 10: a Rank correlation between anomalies of IWV and ice concentration tendency. Only significant correlations are plotted. b Rank correlation between anomalies of northward wind and ice concentration tendency. c Rank correlation between anomalies of latent heat flux and ice concentration tendency. d Rank correlation between anomalies of sensible**

**heat flux and ice concentration tendency.**

Next, we examine mean anomalous conditions when extreme moisture anomalies occur for each grid box in the Arctic Ocean (Figure 11). We identify dates with extreme moisture anomalies specific for each grid box and calculate mean anomalies on those dates locally over that grid box. Note that extreme moisture anomalies are daily

while AR events have varied duration, but mean anomalies with extreme moisture anomalies and AR events are the same. When extreme moisture anomalies (approximate AR events) happen over partial sea ice cover, mean ice concentration tendency anomalies are negative almost everywhere in the Arctic Ocean. Correspondingly, mean

latent and sensible heat flux anomalies are positive, and mean northward wind anomalies are positive (except certain locations near land margins). It is noted that the strength of mean extreme moisture anomalies is similar over the entire Arctic Ocean (Supplementary Figure S4). However, sea ice response shows large spatial variations: magnitudes of ice concentration tendency anomalies are much larger in subarctic seas (e.g., the Bering Sea) than in the central Arctic, indicating that partial sea ice cover in subarctic seas is more sensitive to moisture forcing than in the central Arctic.

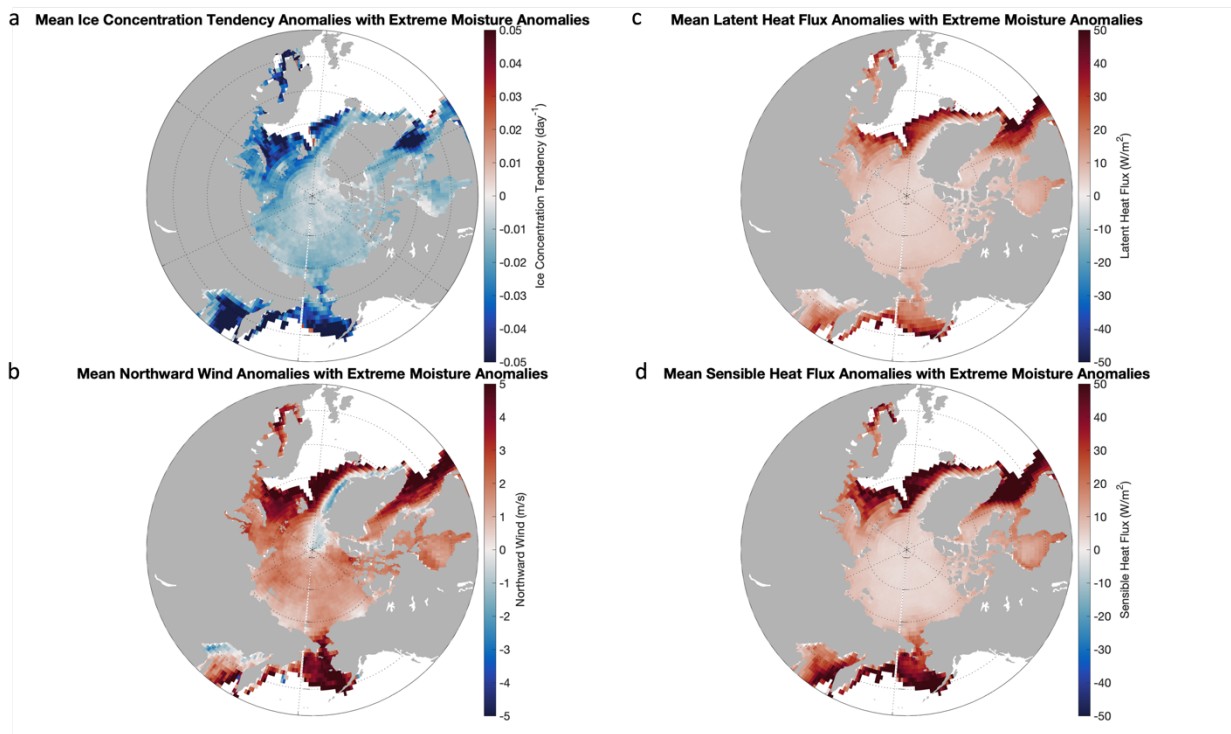

**Figure 11: a Mean ice concentration tendency anomalies with extreme moisture anomalies. b Mean northward wind anomalies with extreme moisture anomalies. c Mean latent heat flux anomalies with extreme moisture anomalies. d Mean sensible heat flux anomalies with extreme moisture anomalies.**

## 4 Conclusion and Discussion

Poleward moisture transport from lower latitudes through ARs has large impacts on Arctic sea ice variations. ARs bring moist and warm air and strong winds into the Arctic. When ARs reach regions covered by sea ice they can trigger rapid and substantial sea ice loss on weather timescales. Sensible and latent heat fluxes are the dominant terms in surface energy budget associated with ARs. These fluxes are enhanced by warm and moist air and strong winds associated with ARs. Due to clouds and precipitation formed by ARs and large atmospheric water vapor content, net longwave radiation is enhanced moderately, and net shortwave radiation is reduced. Additionally, dynamic sea ice motion driven by strong near-surface winds under ARs further reduces sea ice concentration near

ice margins. Note that the relationship between ARs and sea ice loss is similar and even stronger in winter (not shown), though the case studies shown in this paper are in summer.

An analysis over the entire Arctic Ocean shows ice concentration tendency has significant anticorrelation with IWV, northward wind, and turbulent heat flux anomalously almost everywhere. This occurs across seasons as well. The

coherence over the entire Arctic Ocean of anticorrelation between anomalous moisture/winds and sea ice variations on weather timescales provides important implications for Arctic sea ice prediction. The documented anomalies and statistical relationships found in this work helps explain the physics governing the observed sea ice variability associated with ARs. In particular, ARs are characterized by a low-level jet and this study demonstrates the important role that associated with winds play in driving sea ice motion and thus sea ice variability.


Our conclusions are in general consistent with the dominant role of atmospheric temperature in driving sea ice change mainly through advection at low levels (Li et al., 2014 a b; Olonscheck et al., 2019), though over different time scales. Our study shows that near-surface atmospheric temperature and moisture induced sensible and latent heat fluxes from ARs to sea ice are most important terms in surface energy budget on weather timescales

(Olonscheck et al., 2019). ARs large impact on sea ice variations on weather timescales also suggests that ARs provides one mechanism of bottom amplified warming with important implications for Arctic amplification (e.g. Woods and Caballero, 2016).

Though we have added novel analysis regarding the interaction between ARs and sea ice, there is still much to do.

For one, the role of the ocean and ocean feedbacks should be examined, possibly using coupled atmosphere-ocean-sea ice models. Remote sensing is challenging in the presence of ARs, but nevertheless future work could also aim to incorporate satellite inferred sea ice velocity into the type of analysis presented here. Moreover, while we have shown that individual AR events can have large impacts on the Arctic sea ice variations in the short term, the integrated effect of ARs including frequency and strength on the Arctic sea ice budget deserves further study. It is

important to understand the extent to which the Arctic sea ice decline can be attributed to AR activity, possibly using climate models.

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

**Supplementary**

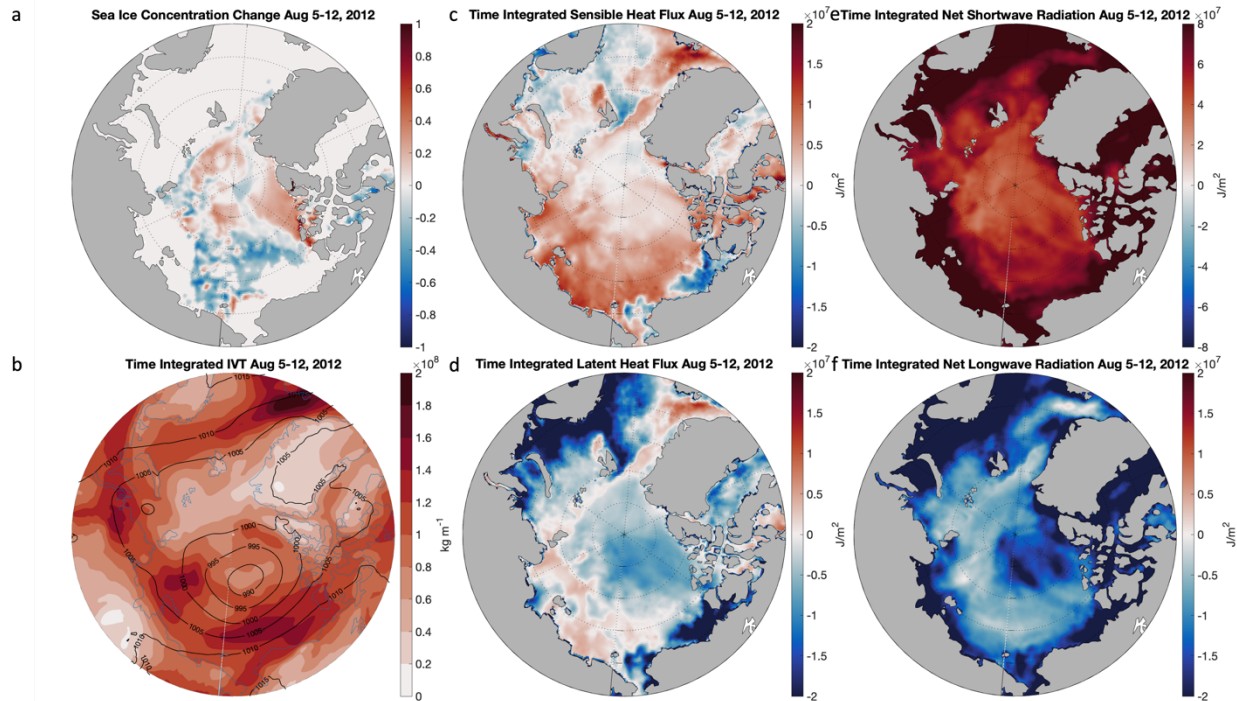

**Figure S1: a** Sea ice concentration change during Aug 5-12, 2012, based on satellite observation. **b** Time integrated magnitude of integrated water vapor transport during Aug 5-12, 2012. Time mean sea level pressure during Aug 5-12, 2012 is represented as black contours. **c** Time integrated sensible heat flux (positive downward) during Aug 5-12, 2012. **d** Time integrated latent heat flux (positive downward) during Aug 5-12, 2012. **e** Time integrated net shortwave radiation (positive downward) during Aug 5-12, 2012. **f** Time integrated net longwave radiation (positive downward) during Aug 5-12, 2012. Time is in UTC.


## Sea Surface Temperature Change Aug 5-12, 2012

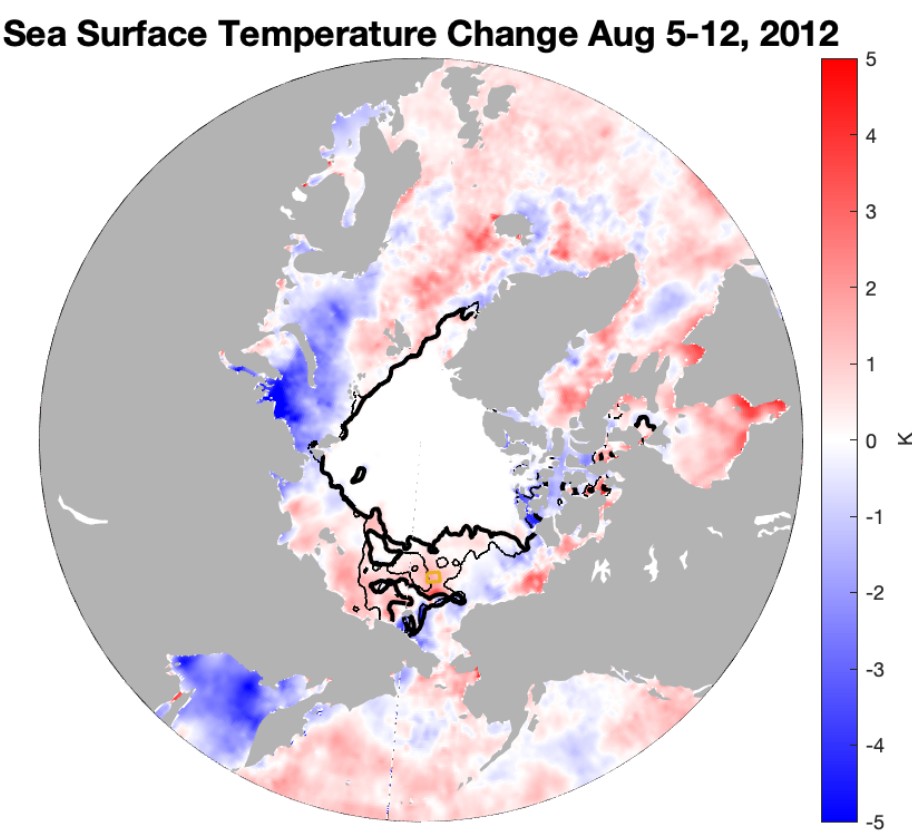


**Figure S2: Sea surface temperature change during Aug 5-12, 2012. Thin black lines represent sea ice edge (15% contour of sea surface temperature) on Aug 5, 2012 and thick black lines represent sea ice edge on Aug 12, 2012. Time is in UTC.**




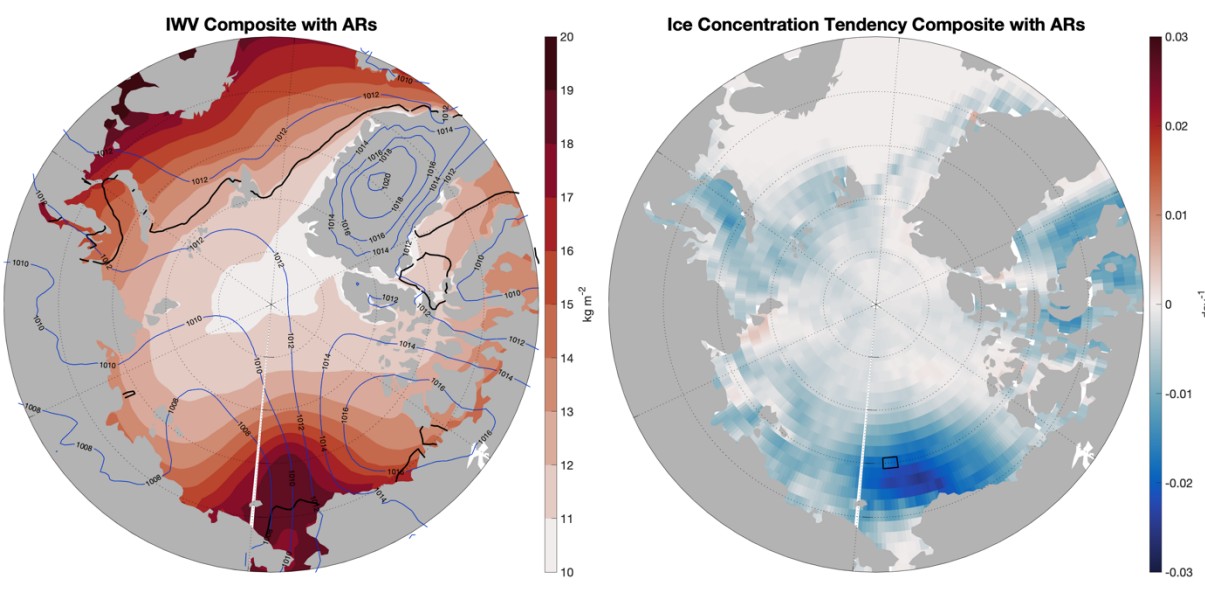


**Figure S3: Composites of IWV, sea ice edge (black lines), sea level pressure (blue contours) and ice concentration tendency with ARs from AR catalog among extreme IWV anomalies over partial ice cover in the black box in the Chukchi Sea.**


# Mean IWV Anomalies with Extreme Moisture Anomalies

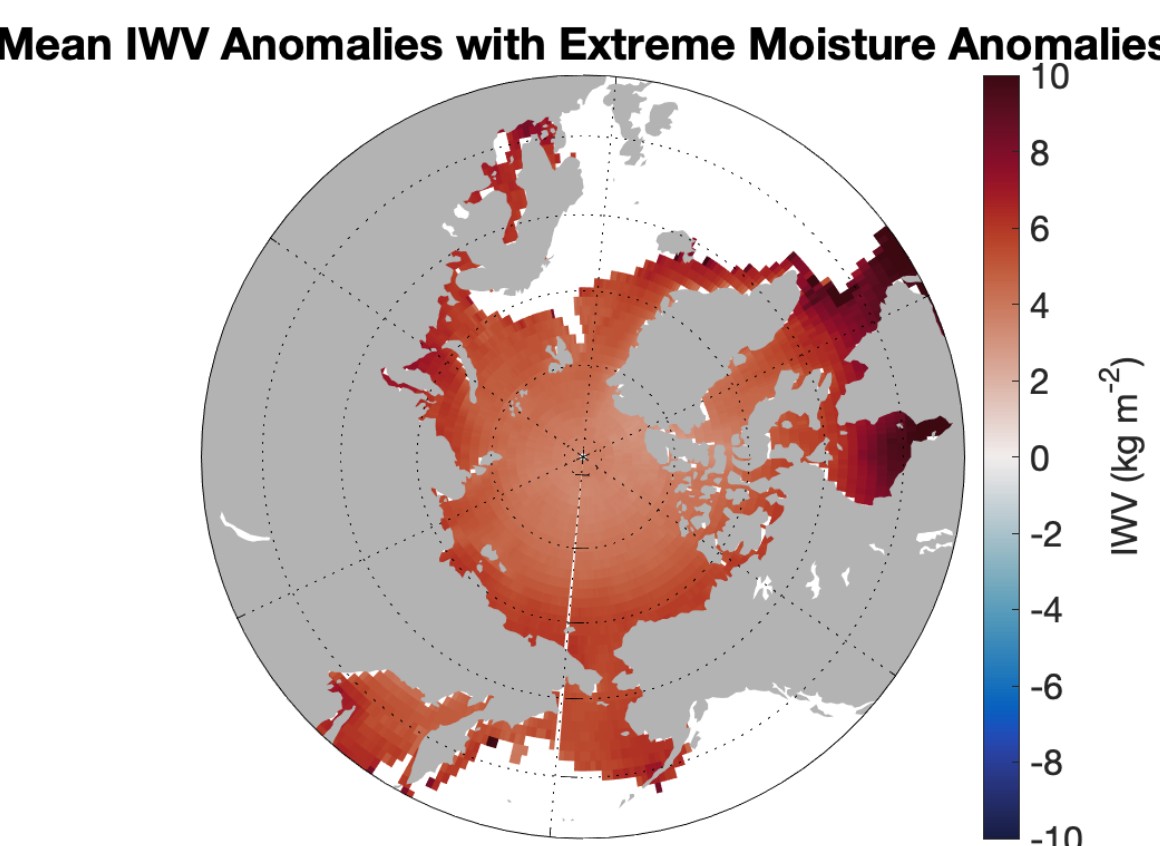

**Figure S4: Mean IWV anomalies with extreme moisture anomalies.**