# Peer review of "Impact of Atmospheric Rivers on Arctic Sea Ice Variations"

_EGUsphere, 2022_

## Author Response (AR1)

**Response to Reviewer 1's Comments**

We thank the reviewer for detailed comments and insightful suggestions. This study has been greatly strengthened and improved by this review. We will revise the manuscript to address each of these comments. Below are our responses to each comment.

*Comments on TC manuscript egusphere-2022-36:*
*This manuscript investigates the impact of atmospheric rivers (ARs) on partial sea ice concentration variation based on observations. The authors started with two cases in the Chukchi Sea in the summers of 2012 and 2020. The surface heat fluxes when ARs approach the ice cover are analyzed. Then, the authors expanded their analysis to the partial sea ice cover in the whole Arctic.*
*General comments:*
*The research is interesting and within the scope of TC. However, I do have some major concerns (Note that this manuscript is revisable). First, the authors state previous literature which addresses results similar to what they are presenting. For general readers, it is hard to see how this manuscript is a significant scientific advance in knowledge. I do think that there are some novel insights in this work, but the novelty should be discussed in a more explicit way. Second, the case study is somehow superficial. I do believe this study could be much in-depth. I have some specific suggestions which may be helpful to improve this study (see details below). Overall, a major revision is needed to make the manuscript publication worthy on The Cryosphere.*
*Specific comments:*

1. *As mentioned above, several previous studies, such as Woods et al. (2016), Hegyi et al. (2018), and others discussed in the Introduction, have reported that the extreme water vapor transport, including ARs, can lead to the Arctic sea ice melt through the surface fluxes (radiative and turbulent) based on case study and statistical correlation analysis. These conclusions and the analysis method are generally similar to the current study. Therefore, I think the authors need to identify how this manuscript is unique from the existing studies, especially what gap of knowledge this manuscript addresses. To my understanding, most previous studies about the role of moisture transport in sea ice melt focused on the wintertime and the Atlantic sector; while the cases studied in the current paper occurred in the summer the Chukchi Sea. To distinguish from existing studies, the authors may confine their analysis to the summer. The surface energy balance in the summer Arctic is very different from winter, so I think focusing on summer (mainly on the Pacific side, including the Chukchi Sea) can be regarded as a novelty. Another suggestion to polish the novelties can be seen in #3.*

Novelties of this manuscript:

1. This study covers all seasons for the entire Arctic Ocean as shown in the statistical analysis. The most important result is that moisture and wind components of ARs correlate well with partial sea ice concentration change on a daily timescale throughout the year almost everywhere in the Arctic Ocean. Though we only show two case studies

during summertime in the Chukchi Sea of the Arctic Ocean, we also examined AR events during wintertime in the Bering Sea with similar effects on sea ice tendency (not included in this manuscript). Therefore, the homogeneous spatial pattern of correlation of AR conditions and sea ice changes and similarity across different seasons are important novelties of this study.

1. ARs are one form of extreme moisture transport operating on weather timescales. Previous studies on moisture intrusion into the Arctic emphasize the important role of incoming longwave radiation. Our results show that turbulent heat fluxes are the dominant terms of surface energy balance; net longwave radiation is only moderate, suggesting that incoming longwave radiation is largely canceled out by outgoing longwave radiation.

1. We explicitly separate the thermodynamic and dynamic effects of ARs on sea ice changes by partitioning moisture and wind components of ARs. Previous relevant studies mostly examine surface energy budget due to the moisture component of ARs. Our study includes the important roles of winds in driving sea ice motion, as ARs are also characterized by a low-level jet. Therefore, our study of ARs has a more comprehensive understanding of the physical processes of the interaction between ARs and sea ice.

1. We develop a new method for detecting extreme moisture anomalies. Our method extracts high-frequency variations on weather timescales, removing seasonal cycles varying year to year. Previous methods remove a fixed seasonal cycle to extract anomalies, which include contributions of interannual variability and weather events. Our method is more suitable for studying extreme weather events such as ARs.

1. *Analysis method:*
*For the case study in Section 3.1, I suggest the authors present the daily anomalies (the departure from the daily climatology or the anomalies used in Section 3.2). Based on the analysis of original values of the surface energy fluxes, the authors argue that sensible heating is dominant. However, with the seasonal climatology in heat flux data, the conclusion might not hold. For example, given the climatological net longwave radiation in summer is negative (downward positive), the weak positive values shown in the figures may indicate a large positive anomaly. Therefore, using the original values cannot tell us which process is the dominant one. The current figures showing the original values can be moved to the supplementary file if the authors want to keep them. In addition, the authors may show how large the magnitude of the anomaly is (for example, exceeding its 1.5/2 standard deviation or not). In this case, Section 3.2.1 and Fig.5 can be removed to save space.*
We will include analysis of daily anomalies of surface heat flux terms for the two case studies in the revised manuscript. Because the definition of ARs is based on the original values of IVT, we start with presentation of original values to quantify the relative importance of each term in surface energy budget for the case studies. However, our statistical analysis over the entire Arctic for 40 years is based on anomalies.

1. *The role of dynamical ice motion, i.e., ice drifting due to the southerly wind associated with ARs, needs further analysis. For a specific location, such as the small box the authors chose, I agree that the wind anomaly could contribute to the local changes in partial sea ice concentration. Considering that ice drifting is regarded as one of the key conclusions in this study, the authors should clearly show the amount of ice drifting rather than the southerly wind only. The ice drifting has been discussed in many cyclone studies but usually has been neglected in AR or extreme moisture transport studies (focusing on thermodynamical effects). Therefore, the analysis of ice drifting (if the evidence shows that it indeed matters) could be regarded as another highlight.*

Sea ice velocity is an important variable in sea ice balance. However, ERA5 does not provide sea ice velocity. We use northward wind as the driving force to approximately study ARs' dynamic effect on sea ice. The partial sea ice cover with ice concentration < 85% we consider is in free drift with small internal ice stress (Heorton et al., 2019). Sea ice motion in free drift is largely driven by wind stress. Sea ice velocity in free drift is consistent with wind forcing especially on weather timescales: ~2% in magnitude, ~30 degree of rotation in direction. Northward wind generally induces sea ice reduction due to divergent sea ice motion near the sea ice edge. Satellite observations and model outputs of sea ice velocity will be examined in future study.

*Technical*
1. *Rank correlation: I might miss some key information here, however, the authors may state how to rank the data. Does the result sensitive to the ranking?*

Spearman's rank correlation is the Pearson correlation between ranks of variables, considering monotonic relationships between variables. Rank correlation is nonparametric and is robust to extreme values.

1. *Some figures should be refined. For example, it is hard for readers to identify the digits in Fig.1b&3b.*

We will improve the figures in the revised manuscript.

1. *Figures 1a & 3a: The authors may clearly state how they define the SIC change during Aug 4-6, 2012.*

Sea ice concentration change during Aug 4-6, 2012 equals to sea ice concentration on Aug 6, 2012 minus sea ice concentration on Aug 4, 2012 (Figure 1a). The figure caption will be revised.

1. *Line 174: the details of the citation are needed. Is it obtained from an NSIDC webpage?*

This will be changed to be 'We study another extreme event in summer of 2020. 2020 experienced the second lowest summer sea ice extent and the lowest sea ice extent during spring, early summer, and fall in the Arctic, based on NSIDC sea ice index (Fetterer et al., 2017).'

1. *Line 208-209: More evidence is needed to support the argument "moisture content is more important than wind speed".*

This is to be changed to 'For this AR event on July 27, 2020, peaking moisture content, along with high wind speed, generates peaking downward turbulent heat fluxes.'

1. *Line 267-268: 15% is conventionally regarded as the edge of sea ice cover. I'm just wondering why the authors chose 85% as the upper limit. Is it an empirical choice? If so, the authors may remind the readers here and state that the conclusion is not sensitive to the choice after test.*

We consider partial sea ice cover with ice concentration between 15% and 85%. This is in general consistent with the definition of the marginal ice zone having ice concentration between 15% and 80%. Also, sea ice is in free drift with small internal ice stress for ice concentration less than 85% (Heorton et al., 2019).

1. *Line 290-291: I'm wondering about the persistence of these events (say the continuous days exceed 1.5 or 2 sigma or 90% percentile). For example, if the mean persistence is 5-day, can we infer that the mean SIC melt per event is -5% at a given location? It would be great if the authors can add a figure to present the timescale of these events.*

Duration of AR events is another important consideration. The case studies show that those two major AR events last less than 1 day (18 hours and 16 hours), based on hourly time series (Figure 2 and Figure 4). In the statistical analysis, we use daily extreme moisture anomalies, consistent with daily satellite observations of sea ice concentration. The regular and consistent time intervals are necessary for calculating correlations. Responding to the specific question the reviewer asks, the mean ice concentration tendency anomalies with daily extreme moisture anomalies are the same as the mean ice concentration tendency anomalies with one AR event lasting for several days, because ice concentration tendency is ice concentration change divided by duration. Duration and other characteristics (e.g. frequency and intensity) of ARs in the Arctic contrasting with ARs at midlatitudes is a very interesting topic for future study.

1. *Line 295, 308-310: Most of the content of Section 3.2 is based on extreme moisture events. I fully understand that ARs and the 90% percentile moisture extremes share similarities and overlaps. Since the topic of this manuscript is AR, showing the analysis using AR (like Appendix Fig.3) would be more consistent, right?*

We develop a new method to extract high-frequency variations on weather timescales and to identify extreme moisture anomalies as approximate ARs which are validated by Guan and Waliser's AR catalog version 3. Our method of detecting daily extreme moisture anomalies is simple and efficient for large datasets from ERA5 and agrees well with the AR catalog from 6-hourly MERRA2. As far as we know, there is no AR catalog based on hourly ERA5, and detecting ARs from ERA5 using the full criteria from Guan and Waliser could be another project.

1. *Line 322-324: There are at least two effects of southerly wind associated with ARs: dynamically redistributing the ice fraction and transporting the water vapor into the Arctic. Thus, the authors may add "In addition to delivering water vapor into the Arctic" or similar words somewhere.*

Winds associated with ARs have at least three effects on sea ice: 1. Northward winds drive divergent sea ice motion near the sea ice edge to cause sea ice to decrease dynamically. 2. High wind speed enhances turbulent heat fluxes. 3. Northward winds transport water vapor and heat from lower latitudes into the Arctic. These three effects are mentioned in the results.

Additionally, ocean conditions could respond to high winds related to ARs (such as enhanced ocean mixing) inducing further sea ice melt.

**Response to Reviewer 1's Comments**

We thank the reviewer for assessment and for directing us towards several important future studies that are related to the physical processes analyzed in this manuscript. The manuscript will be greatly improved by this review. We will revise the manuscript to address each of these comments. Below are our responses to each comment.

*This paper uses reanalysis model output (ERA-5) to assess the contribution of atmospheric rivers (strong advection of water vapor; AR) to melting of ice in the Arctic. The authors describe two episodes of strong AR events in the Arctic and map the various heat flux terms associated with each event. They also plot time series of the contribution of various terms over a span of days centered on those events. A statistical analysis of heat flux terms over a multiyear period is used to suggest the spatial pattern of association between AR activity and ice loss.*

*Although the authors do show some correspondence between AR events and ice loss, I got the impression that these AR events were rare, and that shortwave input was by far the dominant term in the net heat flux melting ice (as an aside, I assume the shortwave term refers to input after reflection by the ice, but the authors need to make this explicit). This is not to say that ARs do not melt ice, and potentially advect it northwards - they authors clearly implicate this mechanism in their two case studies. What is less apparent is whether a year with a few strong ARs would necessarily yield less ice than a year without them. The use of rank correlations, as opposed to standard Pearson's r, is similarly unconvincing to this reader.*

*Similarly, the authors suggest at several points in the text that enhanced net downward longwave radiation is an important contributor to the ice loss - yet the contributions shown in Figures 1 and 2 are exceedingly small.*

One of the new results of this paper is that sensible and latent heat fluxes are the dominant terms in surface energy budget when AR reaches sea ice surface, while net longwave radiation is moderate, based on ERA5 reanalysis data. In contrast, net shortwave radiation is reduced due to clouds and precipitation when ARs happen (Figure 1 and Figure 3), and hourly time series of two case studies show that those ARs happen at midnight when shortwave radiation reaches minimum (Figure 2 and Figure 4). This does not contradict the fact that shortwave radiation dominates the summertime surface energy budget over longer timescales than weather. Our results on the surface energy budget of sea ice are generally consistent with in situ observations (Tjernström et al., 2015; Tjernström et al., 2019) and coupled atmosphere/ocean/ice models (Stern et al., 2020) showing the dominant role of turbulent heat fluxes under ARs.

We use Spearman's rank correlation as it is a non-parametric test, not assuming a normal distribution, and is robust to outliers. The variables (e.g. IWV and ice concentration tendency in Figure 8a) being considered are in fact not normally distributed and are characterized by having extreme values. These extreme values are the focus of this study. Rank correlation assesses monotonic relationships without assuming linear relationships between variables.

The reviewer raises an interesting question of whether a year with a few strong ARs would necessarily yield less ice than a year without them. This deserves more comprehensive study in the future. From our perspectives, many factors contribute to the Arctic sea ice variations. For instance, warm air temperature in the Arctic is likely to be associated with low sea ice years such as 2020. AR is only one possible mechanism causing the Arctic sea ice loss in the short term.

*In summary: while I appreciate the authors attempt to quantify the contribution of ARs to melting ice (i.e. this is an important issue), I do not think they have made a convincing case that such events are big contributors to the ice budget of the Arctic.*

This paper focuses on how ARs interact with the Arctic sea ice thermodynamically and dynamically. The results show that when ARs happen they can cause rapid and substantial sea ice loss on weather timescales. However, the total contribution of all AR events to the Arctic sea ice budget is another important topic for future research. In order to quantify the integrated effect of ARs on the Arctic sea ice, we need to consider the frequency and intensity of ARs. Though individual AR events can have large impacts on sea ice in the short term, the occurrence of ARs over partial sea ice is rare. For example, for the study area in the Chukchi Sea, we identify 553 approximate AR events with 20% frequency (percentage of time of occurrence) when sea ice cover is partial during 1981-2020. Furthermore, other important factors such as the ocean's roles need to be considered to quantify the relative importance of atmospheric conditions related to ARs in the Arctic sea ice budget possibly using coupled atmosphere/ocean/ice models. The introduction and conclusions of the manuscript have been revised to incorporate these important considerations.

*More detailed comments are as follows:*

*l.16 - "longwave radiation" - not shown to be a big contributor*
The contribution of net longwave radiation is only moderate in the surface energy budget.

*l.70 - change "estimate of Arctic" to "estimates of the Arctic"*
Done.

*l.71 - change "near Arctic" to "near the Arctic"*
Done.

*l.74 change "2" to "two"*
Done.

*l.79 change "timescales and to" to "timescales to"*
Done.

*l.90 change "2" to "two"*
Done.

*l.110 This important point regarding the relative magnitude of the fluxes (net longwave being *much* weaker that turbulent flux) needs to reflected in the Abstract, which had implied that longwave flux was important.*
Abstract has been revised accordingly.

*l.115 change "important" to "an important"*
Done.

*Figure 2 - need to specify whether GMT or local time is plotted, and whether the dates on the axis are centered on midnight or noon*
It is GMT, and the dates marked on the axis are at midnight. Figure captions are revised to include time standard.

*Figure 2c - a plot of wind vectors would be more revealing (would show both direction and magnitude, illustrating northward winds)*
We examine wind direction (not shown) and discuss the key results in the manuscript. In two case studies, wind direction is northward when the AR happens and turns eastward after the AR.

*l.152-153 "moisture and wind are both important in contributing to the strong IVT". This statement seems rather circular, since the IVT is basically defined as moisture times wind. Are the authors trying to make the point that the *variance* of the the IVT signal is due equally to both elements?*
This is changed to be ' In summary, simultaneous peaks in moisture and wind speed cause intense downward turbulent heat fluxes and subsequent rapid sea ice decrease when the AR arrives on Aug 5, 2012.'

*l.189 - "southerly" and "westerly" - for consistency, use "northward" and "eastward", and is done earlier in the text*
Done.

*l.209 - "moisture content is more important than wind speed in strong downward surface heat fluxes and raid sea ice decrease" - I don't see this demonstrated in Figure 4; instead, I see a period of high wind speed associated with a steady loss in sea ice concentration.*
We mainly consider the AR event on July 27, 2020, not the cyclone lasting for several days. This is changed to be 'For this AR event on July 27, 2020, peaking moisture content, along with high wind speed, generates peaking downward turbulent heat fluxes.'

*l.226 - "longwave radiation" - I do not see a strong influence of longwave radiation in the shaded plots or line graphs. It's contribution appears to be minor.*
The contribution of net longwave radiation is moderate in the surface energy budget.

*Section 3.2.2 - what is the justification for using rank correlations, as opposed to Pearson's r? The existence of a rank correlation, by itself, does not make a very convincing case for the importance of a forcing term.*
Spearman's rank correlation is non-parametric (not assuming normal distributions) and is robust to extreme values. We also tried Pearson's correlation, which is similar to but slightly weaker than Spearman's rank correlation.

*Figs 8 and 9 - I think it would be far more convincing to show the Pearson's r correlation of ice loss with northward IVT - has this been attempted?*
We explicitly partition moisture and wind components of ARs to have a better understanding of thermodynamic and dynamic processes for a broader community. Using IVT in correlation is an important next step in quantifying the relationships between ARs and sea ice loss.

---

## Referee Report (RR1)

The authors addressed the comments from the reviewers in the last round of revisions adequately. I only have a few comments included below.

General Comments:

Section 4 Conclusions and Future Work: The authors discuss which terms in the surface energy budget change the most under the influence of atmospheric rivers along the ice edge and over the summer sea ice. While these statements fit the results shown in the manuscript, I suggest that the authors are clear that these are likely most important in summer. Both case studies shown in the manuscript are from summer, when shortwave fluxes are higher and conditions near the ice edge are different than in winter.

Specific Comments:

Figure 4a. It is difficult to see the box where the values in Figure 6 are taken from. Is the box along the sea ice edge?

---

## Author Response (AR2)

**Response to Reviewer 1's Comments**

We appreciate the reviewer's positive assessment of this study. Below are our responses to the comments.

*The authors addressed the comments from the reviewers in the last round of revisions adequately. I only have a few comments included below.*

*General Comments: Section 4 Conclusions and Future Work: The authors discuss which terms in the surface energy budget change the most under the influence of atmospheric rivers along the ice edge and over the summer sea ice. While these statements fit the results shown in the manuscript, I suggest that the authors are clear that these are likely most important in summer. Both case studies shown in the manuscript are from summer, when shortwave fluxes are higher and conditions near the ice edge are different than in winter.*

The case studies shown in the first half of this manuscript are in summer. We also examine the winter season and find that the relationship between ARs and sea ice change is similar and even stronger in winter (not shown in this manuscript but presented in 2021 AGU fall meeting). In the second half of this manuscript, we investigate the full daily time series for 1981-2020 everywhere in the Arctic Ocean focusing on weather timescales and find that similar relationship holds for the whole Arctic Ocean through the entire seasonal cycle (Figure 10).

*Specific Comments: Figure 4a. It is difficult to see the box where the values in Figure 6 are taken from. Is the box along the sea ice edge?*

Yes, the box is near the summer sea ice edge. It was partially covered by sea ice before the cyclone and became ice free after the cyclone.

**Response to Reviewer 2's Comments**

We thank the reviewer for detailed comments and insightful suggestions. This study has been greatly strengthened and improved by this review. We revise the manuscript carefully to address these comments. Below are our responses to each comment.

*Comments to "Impact of Atmospheric Rivers on Arctic Sea Ice Variations" by Li et al*

*General comments:*

*This study shows that atmospheric rivers impact the variability of partial sea ice concentration, both in specific case studies and generalized for the Arctic domain. The manuscript is structured*

*well, and explain the physical links and processes that lead from northward moisture transport to reductions in sea ice concentration. To do so, the authors use ERA5 reanalysis data.*

*Although I appreciate the author' improvements based on the previous round of revisions, I still find that the manuscript lacks scientific depth, in particular with respect to how rigorously the authors present their results and the low degree of discussing their findings in the context of previous literature. Not all of there findings are new and they certainly need to be contextualized to what is known about moisture transport to the Arctic and related extreme events. This should appear in the introduction and discussion sections in much more depth. Further, because the authors look at short time scales (days) and primarily on small regions only, the impact of internal climate variability (which atmospheric rivers are part of) is large and hence should be discussed is a broader sense as well. I strongly encourage the authors to properly account for these aspects before I can recommend publication. Relevant literature to start with is e.g.: 1) https://doi.org/10.5194/wcd-3-1-2022, 2) https://doi.org/10.1038/s41561-017-0041-0, 3) https://doi.org/10.1038/s41561-019-0363-1*
*To make that clear, I do not expect just occasional changes, but a substantial rewrite/expansion of at least introduction and discussion.*

We appreciate reviewer's comments on fitting this study in the context of previous studies on moisture transport into the Arctic. We add more references including 3 references suggested by the reviewer. We improve introduction and discussion by summarizing previous relevant studies and emphasizing the novelties of this study.

This study focuses on ARs, which are extreme and episodic moisture transport events on weather timescales. In the general study for the whole Arctic during 1981-2020, we examine the full daily time series for 40 years for each grid box in the Arctic Ocean. Our method of extracting the weather signal from full time series involves removing climatology, interannual variability and trends. Moisture transport on larger timescales, such as interannual variability related to internal climate variability, and trends related to climate change, have very different physical mechanisms and is out of scope of this study.

*Specific comments:*

*1) 119: references for "most former studies" are missing*

References are added.

*2) It becomes not clear to the reader how Supplementary Fig. 1 differs from Fig. 1 and Fig. 2 are for ERA5 and S1 is for satellite observations. Is the point the authors want to make that both look similar?*

Fig 1 uses original fields from ERA5 on Aug 5, 2012 when ARs happened; Fig2 calculates anomalies (original fields minus climatologies) based on ERA5 on Aug 5, 2012; S1 calculates integration of original fields from ERA5 for Aug 5-12, 2012 through the life cycle of the cyclone.

Fig 1 and Fig 2 are similar in showing ARs impacts on surface energy balance, but Fig 2 strengthens the conclusion by removing the climatology. S1 examines the total effect of the cyclone on sea ice. Note that the AR lasts for less than one day, while the cyclone lasts for one week.

*3) l. 171: It is not clear what AR2 means, this needs to be mentioned here.*

This is AR scale determined based on AR duration and maximum IVT during AR (Ralph et al., 2019). AR2 means AR Cat 2 (Moderate): Mostly beneficial, but also somewhat hazardous.

*4) The x-axis labeling in Fig. 3 and Fig. 6 should have the same style.*

The first cyclone happened during August 5-12, 2012 in one month in Fig. 3. The second cyclone happened during July 25 - Aug 2, 2020 in two months in Fig. 6.

*5) The panel titles should be shortened, esp. in Figs. 7, 8, 10, 11.*

---

## Author Response (AR3)

**Response to Editor's Comments**

We appreciate editor's comprehensive review and helpful suggestions for improving this manuscript. The response to editor's comments is below.

*General.*
*\* For clarity, I suggest to use "sea ice" instead of "ice" in every relevant mention... as recently many confusions between "sea ice", "ice sheet" and similar occurred... especially by non-experts. (Optional)*
Done
*\* Provide area (in km^2) for each of the "black boxes", at first mention.*
~100000 km$^2$
*\* Pls be consistent in the use of "i.e.," and "e.g.,", noting fullstops and commas.*
Done
*\* Apply consistent hyphenation, i.e., my preference is, for example, "sea-ice motion". -> Be consistent throughout the manuscript.*
Done
*\* Suggest to add an "Acknowledgement" section for data or information used. Suggest to also acknowledge contributions of (three reviewers).*
Done
*\* Note: References not checked.*
*\* Note: Figure captions not checked.*
*\* Note: Some labels/titles of figures are too small to read.*

*47: Suggest to remove ", which is a primary goal of this paper".* Done
*53: Replace "deserves more studies." with "requires further investigation."* Done
*60: Replace "when ARs happen in the Arctic" with "associated with Arctic ARs".* Done
*60: I understand why you replace "Novel aspects" with "Originality" but recommend to revert to the earlier wording via "A novel aspect".* Done
*61: And replace "includes revealing" with "reveals".* Done
*61: Add "the" to read "and the AR's wind effect".* Done
*61: Remove "change".* Done
*61/62: Replace "on weather timescales" with "at synoptic timescale".* Timescale is weather, and spatial scale is synoptic.
*66: Replace "applications to" with "implication on".* Done
*68: Suggest to replace "influence" to "contribute to".* Done
*69: Replace "and dynamic winds" with "as well as dynamic winds".* Done
*123: Replace "lowest ever observed since satellite observations started in 1979" with "lowest in the satellite record".* Done

*124: Replace "Before that, an Arctic cyclone … (Simmonds and Rudeva, 2012)." with "This was preceded between August 5 - 12, 2012 by a severe Arctic cyclone, one with deepest central pressure (966 hPa) since August 1979 (Simmonds and Rudeva, 2012)."* Done

*126: Replace "AR entered" to "AR arrived in".* Done

*128: Correct "decreases" with "decreased". 129: Correct "enters" with "entered", "reaches" with "reached".* Done

*131: Correct "is" with "was".* Done

*132: Add information on how you inferred the NE surface winds. I.e., "(inferred from sea level pressure from the geostrophic balance)".* Done

*132: Correct "push" with "pushed".* Done

*134: Charge "Coincident wit … (Figure 1cd)." with "This coincided with strong downward sensible and latent heat fluxes near the sea ice edge, driven by warm and moist air and high wind speed at low levels within the AR (Figure 1cd).* Done

*136: Add and correct "near the ice edge is also" to "near the sea ice edge was also".* Done

*138: Remove "cover" to read "over sea ice".* Done

*139: Correct "are" to "were".* Done

*140: Replace "gives" with "provides".* Done

*140: Suggest to replace "can be the" with "ma be the".* Done

*141: Change "ARs." to "an AR."* Done

*141: Change "In contrast, most former… and clouds" to read "This is in contrast to prior studies, which identified dominance of downwelling longwave radiation in relation to water vapor and clouds".* Done

*Figure 1a: The black box is VERY small… need to enhance it… ie with an arrow pointing towards it? -- Also bad to have the "black box" and the "black 15% SIC contour".* The figure will be modified once software can be accessed.

*149: Add note in caption (Fig. 1) about the black box in 1a.* Done

*154: Is this needed? Here in the caption? If not, I suggest to remove "Time is in UTC."* Done

*165ff: Replace "Figure 2… shortwave component." with "In the Chukchi Sea anomalies of the turbulent heat flux during the event collocated with IVT anomalous (Figure 2), whereas significant radiative anomalies occurred throughout the domain especially for the shortwave component."* Done

*175: Shorten by removing "In order", and "also".* Done

*177: Remove "cover" from "sea ice cover" to read "sea ice".* Done

*180ff: Change "The effect … by winds." to read "In the Western Arctic Ocean, the cyclone extensively reduces the sea ice over its life cycle, corresponding to strong IVT, atmospheric warming (enhanced sensible/latent heat fluxes and longwave radiation) as well as by sea ice advection driven by winds."* Done

*185 - 190: This addition seems to contradict itself. Pls streamline and ensure to deliver a clear argument.*

*For example, change to: "While our study focusses on ARs and their effect on Arctic sea ice, the roles of cyclones/anticyclones in sea ice chages is complex and varies between seasons and regions. For example, Wernli and Papritz (2018) showed that enhanced sea ice melt during Arctic summer is related to polar anticyclones as well as extratropical cyclones, however, overall*

*cyclones seem to have less effect on sea ice than long wavelenngth atmospheric waves (Hofsteenge et al., 2022), i.e., those associated with ARs."* Done

*196: Correct "is chosen" to "was chosen".* Done

*197: Correct to read "before the arrival of the cyclone".* Done

*199: Remove "The time series shows that" to read "The sea ice concentration...".* Done

*201: Correct "occurs" to "occurred".* Done

*202: Correct "are related" to "were related".* Done

*203: Correct "enhances" to "enhanced".* Done

*204: Correct "This AR event on Aug 5, 2012 over the black box" to "Over the black box the Aug 5, 2012 AR event".* Done

*206: Correct "separate thermodynamics and dynamics of ARs" to "separate into thermodynamic and dynamic of AR components".* Done

*207: Change "shows a prominent peak" to "peaks prominently".* Done

*211: Clarify by changing "peaks" to "maxima".* Done

*212: Change "when the AR arrives on Aug 5, 2012." to "as the AR arrived on Aug 5, 2012."* Done

*217: Clarify by changing "averaged over" to "area averaged over".* Done

*220: Clarify by changing "averaged over" to "area averaged over".* Done

*221: Not sure if the following information is really needed: "Time is in UTC, and dates marked on time axis represent 00:00 UTC."* Deleted.

*223: Correct "decreases" to "decreased".* Done

*233: Change "We study... (Fetterer et al., 2017)." to "The second lowest sea ice extent minimum (recorded in 2020, based on the NSIDC sea ice index (Fetterer et al., 2017)) is also investigated for its relation to AR forcing."* Done

*237: For consistency change "Jan" to "January".* Done

*268: Pls check contradicting wording "decreases GRADUALLY" when contrasting to Aug 2012.* I think that is the case.

*267: Remove "first" from "the first case".* Done

*289-291: Avoid single sentence paragraph. Consider merging with next paragraph or later discussion.* Merged with next paragraph.

*287-289: "Our future work will examine the relative contribution of water vapor/heat transport and local warming/moistening in sea ice decline in the Arctic." -> Consider moving to the final section (outlook) of this manuscript.* Moved to discussion.

*293: Remove "While".* Done

*294: Change "here we extend" to "which we extend here".* Done

*294: Remove "more" from "more general".* Done

*294: Replace "for 40 years" with "spanning four decades".* Done

*328: Replace "40 years" with "four decades".* Done

*329: Add "the" to read "from the daily time series".* Done

*331: Change "15% and 85%" to "15 and 85%".* I think 15% is more accurate.

*332: Change "15% and 80%" to "15 and 80%".* I think 15% is more accurate.

*332: Add reference for "definition of marginal ice zone".* Done

*339: Provide reference and/or stats method for "all with p-values <0.01".* Done

*340-41: Change "The big red ... 1981-2020." to "The AR event from Aug 5, 2012 (big red dot in Figure 8a) stands out as an extreme event during 1981-2020."* Done

*365 and prev lines: Change "indicates" to a much stronger term, similar replace "good agreement" with "the congruent argument" or some strong term... and make the lines much stronger.* Done

*387: Change "to each grid box" to "to all grid boxes".* Done

*396: Change "15% and 85%" to "15 and 85%".* I think 15% is more accurate.

*426: Typically one would first have the "discussion" followed by the "conclusion. Change to "Discussion and Conclusion".* Done

*432: Change "formed by" to "associated with".* Done

*434: Remove "dynamic" from "dynamic sea-ice motion".* Done

*435: Change "ice margins" to read "the sea-ice margins".* Done

*436: Add "here" to "(not shown)" to read "(not shown here)".* Done

*439: Correct "across seasons as well." to "year-round."* Done